# Spatial-temporal characteristics and influence factors of high-quality development of animal husbandry industry in China

**Tiantian Su**, **Cuixia Li***

College of Economics and Management, Northeast Agricultural University, Harbin, China

* licuixia.883@neau.edu.cn

**Data Availability Statement:** Data are uploaded to OPENICPSR: https://www.openicpsr.org/openicpsr/workspace?goToPath=/openicpsr/211882&goToLevel=project.

## Abstract

The animal husbandry industry stands as a pillar of the agricultural sector. According to the defining characteristics of high-quality development in animal husbandry, this paper establishes a comprehensive evaluative indicator system. This system evaluates the quality of development in the industry from 2010 to 2022, including five key dimensions: output efficiency, product safety, resource conservation, environmental friendliness, and the level of scientific and technological &management. Results indicate a positive overall trend in the high-quality development of China's animal industry throughout this period. Provinces exhibiting the highest average levels of development are, in descending order, Jiangsu, Shanghai, Anhui, Beijing, Fujian, Yunnan, Chongqing, Gansu, Guangdong, and Guangxi. Among these five dimensions, the level of environmental friendliness and the level of scientific and technological &management are better developed, while the level of resource conservation, the level of product safety, and the level of output efficiency are poorly developed. Regional differences have demonstrated a slight narrowing trend; however, the effect of intra-regional differences on the overall differences has remained relatively stable. In contrast, the effect of inter-regional differences on overall differences has gradually reduced, while the effect of hypervariable density has steadily increased, becoming the most significant factor. High-quality development in China's animal industry exhibits a strong path dependency, with advancements typically occurring incrementally between adjacent levels and an insignificant probability of leapfrogging. The level of high-quality development in a given region is significantly influenced by the development levels observed in neighbouring regions, illustrating a phenomenon akin to the adage that One who is near vermilion is stained red, one who is near ink is stained black. Factors such as urbanisation rates, levels of scientific and technological innovation, transport infrastructure, levels of agricultural financial development, and population density all contribute positively to the high-quality development of China's animal industry. At the same time, a larger share of animal husbandry in both the overall industrial sector and the agricultural industry further contributes to this high-quality development; whereas, GDP per capita and fiscal support for agriculture do not appear to have a significant effect.

**Funding:** This work was funded by the National Natural Science Foundation of China (The effect of infant milk powder safety trust index on product competitiveness—Index measurement, Correlation model construction and market simulation, Project number 71673042) Prof Cuixia Li is the fundraiser of the manuscript and has contributed greatly to the writing of this paper.

**Competing interests:** The authors have declared that no competing interests exist

## 1. Introduction

China's animal industry has become a cornerstone of its agricultural sector, representing 34% of agricultural output [1]. This development has been driven by increasing demand for meat, eggs, milk, and other livestock products. The animal industry plays a critical role in stimulating rural economic growth, enhancing farmers' incomes, ensuring national food security, and achieving agricultural and rural modernisation. However, under the pressure of increasing resource constraints, China's animal husbandry industry continues to suffer from low production efficiency, a large feed-grain deficit, severe environmental pollution, limited technological advancement, and the unsustainable utilisation of grassland resources [2–4]. Moreover, the upgrading of China's consumption patterns and rising living standards have fuelled a burgeoning demand for green, high-quality, and safe livestock products, placing greater demands and challenges on the industry. Therefore, conventional livestock production methods have proven inadequate in meeting the population's health, safety, and environmental needs, making the transition from rudimentary practices to an intensive, high-quality development model a necessity [5]. High-quality development in animal husbandry represents a crucial strategic programme introduced by China in its new phase of development. This programme proposes: innovation to provide impetus for development, coordination to solve the problem of unbalanced development, green to solve the problem of harmonious coexistence between human beings and nature, openness to solve the problem of internal and external linkage of the country, and sharing to solve the problem of social harmony [6]. In 2020, the General Office of the State Council released the "Opinions on Promoting the High-Quality Development of the Animal Husbandry Industry, "(https://www.gov.cn/gongbao/content/2020/content_5551804.htm) advocating for the advancement of the sector towards a new paradigm characterised by high-efficiency outputs, product safety, resource conservation, environmental friendliness, and effective governance [7]. In addition, in 2023, the State also amended the Livestock Law of the People's Republic of China(https://nynct.guizhou.gov.cn/ztzl/wsdwbsgkdwpd/gzscqzyglz/zcfg_41025/202312/t20231219_83370442.html) in terms of doing a good job of harmless treatment of livestock and poultry manure, promoting the balance of grass and livestock, and strengthening the green development of the livestock industry, aiming to regulate the production and operation of the livestock industry and promote the high-quality development of the livestock industry. So, what are the characteristics of high-quality development of China's animal industry in time and space? What are the characteristics of regional differences? What factors influence the level of high-quality development in China's animal industry? By quantitatively analysing these issues based on a scientifically established evaluation index system for high-quality animal husbandry in China, this paper aims to offer valuable theoretical insights to promote high-quality development of animal husbandry industry in China.

At present, most scholars have focused mainly on studies related to the level of high-quality development in agriculture.; whereas, analyses concerning high-quality development in the animal husbandry sector remain limited. From the perspective of qualitative aspect, the scholars mainly summarise the achievements and problems faced by China's animal husbandry industry, the connotation of high-quality development of animal husbandry and the path to achieve it. Since the reform and opening up, China's animal husbandry industry has made great achievements. Productivity has risen dramatically, with per capita meat, egg and milk consumption already at 140. 2%, 213. 2% and 28. 0% of the world average, respectively [8]. The structure of the industry is increasingly optimised, with the proportion of beef, mutton and poultry meat in meat consumption gradually increasing. The level of scale is steadily increasing, with the scale of livestock and poultry farming reaching 67. 5% in 2020 [9]. The support and guarantee system is also improving day by day [9]. However, the development of animal husbandry in China is

still facing many challenges, such as tighter resource constraints, stricter environmental protection constraints, more severe epidemic prevention and control, high degree of external dependence on core technologies and weak international competitiveness [8,9]. High-quality development of the animal industry is an important strategic initiative to meet the challenges, and its connotation has been interpreted by different scholars from different perspectives. Generally speaking, it is to achieve high quality in the production process and results [10]. The high quality of the production process is to take green development as the core, science and technology as the driving force, a safe and sufficient forage system and a high degree of attention to animal welfare as the breeding environment, and large-scale, standardised and industrialised as the mode of production. A paradigm of intensive livestock production that is highly productive, economically efficient, highly resource conservation, and environmentally friendly will eventually be developed. The high quality of the production results is to meet the comprehensive, diversified and personalised demand of the residents for livestock products, and to provide the residents with safe and high-quality livestock products [8–12].

From the perspective of the quantitative aspect, scholars mainly focus on the evaluation of the development of animal livestock modernisation and the evolution of spatio-temporal dynamics. In the evaluation of indicators, different dimensions of indicator construction lead to different measurement results. The scholars mainly focus on five dimensions of material and equipment level, scientific and technological progress level, operation and management level, safe and sustainable level, and industrial output level [13];five dimensions of green development, industrial output, scientific and technological progress, material and equipment, and operation and management level [14];four dimensions based on the industrial chain perspective from the upstream, midstream, downstream, and resourcefulness utilization [15];and four dimensions of resource conservation, environmental friendliness, product safety, and output efficiency [16] to construct the index system of animal husbandry modernisation or green development level. In terms of spatial and temporal evolution characteristics, the Gini coefficient and kernel density maps are mainly used to reveal their dynamic evolution characteristics. In terms of high-quality development of the animal industry, only Xiong Xuezhen (2022) [17]measured the level of high- quality development of the animal industry from three dimensions: green cycle development, supply quality and efficiency, and optimisation of business management. Few scholars have explored the influencing factors on the high-quality development of animal husbandry, and most scholars have explored the influence of urbanisation rate, transport level, scientific and technological innovation level, digital economy development level, financial support for agriculture level, environmental regulation, industrial structure, rural finance and so on [18–22]on the high-quality development of agriculture, which provides theoretical references to the research of this paper.

In summary, drawing upon the requirements for high-quality development of the animal industry in the "Opinions on Promoting High-Quality Development of Animal Husbandry, "this paper proposes a novel method for evaluating the high-quality development of this industry in China, comprising five key dimensions:high output efficiency, product safety, resource conservation, environmental sustainability, and level of scientific and technological &management. It is important to note that due to the challenges in obtaining data related to the regulation of effective indicators, the fifth dimension of this paper focuses specifically on measuring the level of existing science, technology and management practices in China's animal industry. The entropy weight method and cluster analysis are used to measure and rank the level of high-quality development of China's animal husbandry industry from 2010 to 2022. On this basis, the Dagum Gini coefficient is used to analyse the regional differences in the high-quality development of China's animal industry as well as the sources, and Kernel density estimation, Markov chain and social network models are used to simulate the dynamic evolution

characteristics of China's high-quality development of the animal industry and the characteristics of the spatial correlation network. Finally, the multiple linear regression method reveals the influencing factors of the high-quality development of China's animal husbandry industry, and puts forward some countermeasure suggestions for accelerating the promotion of the high-quality development of China's animal husbandry industry.

## Materials and methods

### 2. 1. Research methods

**2. 1. 1. Entropy weight comprehensive evaluation method.** The entropy weight method offers a comprehensive means of evaluating situations characterised by multiple indicators and characteristics. The method carries out comprehensive evaluation by considering the correlation and importance between indicators and determining the weight of each indicator. The entropy weight method can be broken down into the following steps:

1. Indicator Identification and Data Acquisition: Begin by identifying the specific indicators requiring evaluation and collect the relevant data.

2. Data Standardisation: Standardise the data for each indicator to ensure comparability across the dataset.

3. Entropy Value Calculation: Calculate the entropy value for each indicator. This value reflects the information content and uncertainty associated with each indicator.

4. Weight Calculation: Derive the weight of each indicator from its corresponding entropy value. The entropy weight method hypothesizes that indicators with higher entropy values should be assigned greater weights, and vice versa.

5. Comprehensive Evaluation: To arrive at a weighted score for each indicator, multiply its weight by its standardised value. Sum these weighted scores across all indicators to produce the final comprehensive evaluation result.

The entropy weight method is highly effective because it considers the weight and information content of each indicator, circumventing the pitfalls of subjective weight assignment. Therefore, it enjoys widespread application in multi-indicator decision- making and evaluation processes [23–26].

**2. 1. 2. Dagum Gini coefficient and decomposition method.** The Dagum Gini coefficient represents an optimization of the traditional Gini coefficient. Its key advantage is reflected in its decomposability into intra-group coefficient, inter-group coefficient and hyper-variable density coefficient. By addressing the issue of overlapping data, which often limits other regional difference measures, the Dagum Gini coefficient offers superior identification of the root causes driving regional differences. The formula is:

$$G = \frac{\sum_{j=1}^{k} \sum_{h=1}^{k} \sum_{i=1}^{nj} \sum_{r=1}^{nk} |y_{ji} - y_{hr}|}{2n^2\bar{y}} \tag{1}$$

Where k is the number of regional divisions, n represents the number of provinces in each region, $nj$, $nh$ represent the number of provinces in the region j, h. $y_{ji}$, $y_{hr}$ represent the high quality level of animal husbandry in provinces i, r in regions j, h. $\bar{y}$ denotes the average level of high-quality development of China's animal industry. The Dagum Gini coefficient can be decomposed into three components: the contribution of intra-regional differences ($G_W$), the

contribution of inter-regional differences ($G_{nb}$), and the contribution of hypervariable density ($G_t$), and satisfy $G = G_W + G_{nb} + G_t$[27–29].

**2. 1. 3. Kernel density estimation method.**   Kernel density estimation is a non-parametric statistical method used to estimate the probability density function of a random variable. This method estimates the overall probability density function by placing a kernel function (usually a Gaussian kernel function or a rectangular kernel function) around each data point, and then weighting these kernel functions to estimate the overall probability density function, which is effective in observing the dynamic evolution of the sample data[30–32]. Assume that the density function of the random variable x is:

$$f(x) = \frac{1}{Nh}\sum_{I=1}^{N} K\left(\frac{X_i - x}{h}\right) \tag{2}$$

Where, N is the number of observations, $x_i$ denotes the observation value, x is the mean value, K($\cdot$) represents the Kernel density, and the bandwidth is denoted by h. Consistent with most studies, in this paper, Gaussian kernel function is chosen to estimate the distributional dynamics of high quality development of China's animal husbandry industry, see Eq (3)

$$k(x) = \frac{1}{\sqrt{2\pi}}\exp\left(-\frac{x^2}{2}\right) \tag{3}$$

In essence, kernel density estimation offers a flexible and robust statistical method for understanding the distributional characteristics and probability density functions in data.

**2. 1. 4. Markov chain analysis.**   Markov chain analysis, a mathematical tool, is utilised to study stochastic processes. Specifically, it focuses on processes where the probability of transitioning between states depends solely on the present state, independent of past states–a property termed the Markov property. In this analysis, a system can exist in a range of possible states, with the transition probabilities between these states determined by a predefined probability distribution. By modelling and analysing these state transfer probabilities, we can gain a deeper understanding of the dynamic evolution laws and characteristics of China's animal industry's development towards high quality.

$$p\{X(t) = j \mid X(t-1) = i, X(t-2) = i_{t-2}, \ldots, X(0) = i_0\} = P\{X(t) = j \mid X(t-1) = i\} \tag{4}$$

From Eq (4), the state of random variable X at a future time period t is solely influenced by its state one period prior to t, i. e., X has the property of first-order Markov. In addition, the transfer probability $p_{ij}$ of random variable X from state i to state j can be obtained by $n_{ij}/n_i$. *nij* denotes the total number of times state i transitions into state j, and *ni* is the number of times state i occurs. Therefore, by analysing the state dimension matrix P, constructed from these transition probabilities, we can infer the dynamic evolution trends in the distribution of high-quality development in China's animal husbandry industry [33–35].

**2. 1. 5 Multiple linear regression.**   It is necessary to conduct a deeper analysis of the factors influencing the high-quality development of animal husbandry industry to offer insights of how to cultivate high-quality development in the China's animal industry. This paper evaluates these influencing factors from a macro perspective, employing the following analytical model:

$$\begin{aligned} LI_{it} &= \alpha_0 + \alpha_1 PGDP_{it} + \alpha_2 URB_{it} + \alpha_3 TEC_{it} + \alpha_4 TRA_{it} + \alpha_5 IND_{it} + \\ &\quad \alpha_6 FIN_{it} + \alpha_7 ARI_{it} + \alpha_8 POD_{it} + \alpha_9 PFA_{it} + \gamma_i + \mu_t + \rho_{it} \end{aligned} \tag{5}$$

The left-hand side of the equation represents the level of high-quality development in the animal husbandry sector in province i in year t. Among the explanatory variables on the right-

hand side of the equation, PGDP represents GDP(Gross Domestic Product, refers to the final results of production activities of all resident units of a country (or region) in a certain period of time) per capita, while URB represents the urbanisation rate [36]. In addition, TEC denotes the level of scientific and technological development in the animal industry, measured by the sum of agricultural patents due to the absence of separate data specifically for the animal industry [37]. TRA signifies the level of transport, measured by the transport network density in each province and calculated as the ratio of the total number of miles of railways and highways to the area of the administrative divisions of each province [38]. IND represents the industrial structure, measured by the proportion of primary industry in the total sum of primary, secondary, and tertiary industries. FIN denotes the level of agricultural financial development, expressed by the total amount of loans for agriculture, forestry, animal husbandry, and fishery in each province [39]. ARI signifies the structure of the agricultural industry, expressed as the proportion of animal husbandry output value in the total output value of agriculture [40]. POD represents population density, measured by the ratio of the year-end resident population to the area of the administrative divisions in each province at the year's end. Finally, PFA denotes the level of fiscal support for agriculture, expressed as the percentage of local fiscal expenditure on agriculture, forestry, and water affairs in the fiscal general budget expenditures. The variables $\alpha_1 \sim \alpha_6$ are the coefficients to be estimated for each of the influencing factors, while, $\alpha_0, \gamma_i, \mu_t$ and $\rho_{it}$ refer to the intercept term, individual effect, time effect, and random interference term, respectively. The data in this section are primarily sourced from the China Statistical Yearbook and the statistical yearbooks of each province.

## 2. 2. Construction of the indicator system

With Chinese socialism entering a new era, the nation's economic development has transitioned from an emphasis on "quantity" to one on "quality". The proposition of high-quality development in animal husbandry not only speaks to the burgeoning desire of the populace for an improved standard of living, but also represents an essential strategy for addressing the pressures of resource scarcity and environmental degradation. This paper will concentrate on the following five aspects to construct evaluative indicators. See Table 1 for specific indicators.

First, the level of output efficiency. Considering existing resource constraints, improvements to production efficiency are essential if the animal industry are to meet the ever- growing demand for products. For the animal industry, achieving greater output efficiency signifies producing a higher output utilizing the same or even fewer resource inputs. This naturally results in the production of greater quantities of meat, eggs, milk and other valuable livestock products. From an economic perspective, this improvement in output efficiency translates to lower feed costs, labour costs, and other expenditures related to farming. It also leads to the optimisation of resource allocation and the maximisation of economic benefits. This paper selects five key indicators to measure the output efficiency of China's animal husbandry industry:livestock slaughter rate, cost profitability, production efficiency, the level of meat consumption, and the level of milk consumption [13,17,45].

Second, the level of product safety. This signifies that livestock products must be free from harmful substances, remain uncontaminated and unspoiled, adhere to relevant health standards and legal and regulatory requirements, and finally, protect consumer health and safety. More specifically, additives, antibiotic residues, heavy metals, and other harmful substances present in feed will affect the safety of livestock products. The unreasonable utilisation of veterinary drugs, antibiotics, and other medications may lead to excessive residues in animal products, which can, accordingly, harm human health. In addition, the hygienic conditions of the breeding environment are directly correlated with the quality and safety of animal products.

**Table 1. Indicator system for evaluating the level of high-quality development of the animal husbandry industry.**

| Primary indicators | Secondary indicators | Interpretation of indicators | Indicator properties |
|---|---|---|---|
| Level of output efficiency | Livestock slaughter rate | Number of animals slaughtered in the period/Stock at beginning of period | + |
| | cost margin | Net profit/total cost | + |
| | Level of meat consumption | Per capita meat consumption | + |
| | Level of milk consumption | Per capita milk consumption | + |
| | production efficiency | The input indicators are the number of labourers in the animal industry, intermediate consumption in the animal industry, and investment in fixed assets in the animal industry, and the output indicator is the total output value of the animal industry, which is calculated by applying the super-efficiency SBM model, and the details of the calculation methodology can be found in the study by Chun Yang (2019) [41] | + |
| Level of Product safety | the cost rate of epidemic prevention | Epidemic defence cost/production costs | + |
| | Level of green food production | Number of green food certifications | + |
| | the qualified rate of livestock and poultry product quality and safety monitoring | Quantity of qualified products/total quantity of products | + |
| | The qualified rate of feed product quality and safety monitoring | Quantity of qualified products/total quantity of products | + |
| | Share of senior and middle-level technicians in the number of staff on board at township animal husbandry and veterinary stations | Total number of senior and middle-level technicians/Total number of staff on board at livestock and veterinary stations | + |
| Level of resource conservation | Level of intermediate consumption resource use | Gross value of livestock production/Pastoral intermediate consumption | + |
| | Level of utilisation of feed resources | Livestock output/feed industry output | + |
| | Efficiency of feed grain production | The input indicators include the amount of fertiliser applied for grain production, the total power of machinery for grain production, and the area sown for grain, while the output indicator is the total grain output, which is calculated by using the super-efficient SBM model, and the details of the calculation method can be found in the study by Chuanming Liu (2023) [42] | + |
| | Level of utilisation of labour resources | Gross value of livestock production/number of labourers employed in the animal husbandry sector | + |
| | Level of land resource use | Gross value of livestock production/area of pasture land | + |
| Level of environmental friendliness | Level of resource use of livestock and poultry manure | Comprehensive livestock and poultry manure utilisation rate | + |
| | Level of household biogas utilisation | Rural household biogas production/number of rural population | + |
| | Carbon emission levels | Carbon emissions from faecal and gastrointestinal fermentation (calculated based on IPCC coefficients), see Jinxin Zhang (2020) for details of the calculation process. [43] | – |
| | Livestock manure land carrying capacity | Calculated according to the calculation method in the Technical Guidelines for Measuring the Land Carrying Capacity of Livestock and Poultry Manure | + |
| | Green total factor productivity | The input indicators are the number of labourers in the animal industry, intermediate consumption in the animal industry, and fixed asset investment in the animal industry, the desired output indicator is the total output value of the animal industry, and the non-desired output indicator is the carbon emission, which is calculated by using the super-efficiency SBM model, and the method of calculation is detailed in the study by Biao-wen Xu (2019). [44] | + |
| Level of science and technology &management | Mechanical equipment strength | Number of units of livestock (breeding) machinery/total livestock production | + |
| | Degree of livestock and poultry farming scale | Number of large-scale farms (households)/total number of farms (households) | + |
| | Quality of labour force | Average level of education of agricultural labourers | |
| | Percentage of senior staff at livestock stations | Number of senior employees in livestock stations / Total number of employees in animal husbandry stations | + |
| | Percentage of stock of breeding animals | Stock of breeding animals/Total stock of livestock and poultry | + |

**Note:** In this paper, the scale level is determined according to the criteria of annual pig slaughtering≥500, dairy cattle stocking≥100, beef cattle stocking≥100, sheep stocking≥100, egg-laying chicken stocking ≥2000, broiler stocking ≥5000.

Epidemic outbreaks can result in health issues for livestock and poultry, thereby affecting the quality and safety of the products derived from them. Considering the available data, this article utilises the following factors to measure the quality and safety level of the animal husbandry industry [46]:the cost rate of epidemic prevention;the level of green food production;the qualification rate of livestock and poultry product quality and safety monitoring;the qualification rate of feed product quality and safety monitoring;and the proportion of senior and mid-level technicians among the total number of staff at township animal husbandry and veterinary stations [13,45].

Third, the level of resource conservation. Livestock farming necessitates large inputs of resource allocation, including water, land, feed, and energy. Therefore, prioritising resource conservation can reduce consumption without compromising output. Regarding feed, optimisation of formulations and processing methods can enhance utilisation. For water resources, implementing recycling systems such as rainwater harvesting and efficient irrigation is crucial to minimise wastage. Land resources are best managed through intensive farming practices, such as ecological animal husbandry and recycling agriculture, to maximise the utility of finite land. In addition, integrating high- efficiency energy equipment and technologies, including energy-saving light sources and solar energy utilisation, effectively reduces energy consumption throughout the livestock production process. This paper employs five indicators to measure the effectiveness of resource conservation in animal husbandry, contingent upon data availability: the level of intermediate consumption resource use, the level of feed resource use, the efficiency of feed grain production, the level of labour resource use, and the level of land resource use[].

Fourth, the level of environmental friendliness. According to the Second National Pollution Source Census Bulletin, the animal industry discharged 10, 005, 300 tonnes of chemical oxygen demand (COD), 110, 900 tonnes of ammoniacal nitrogen, 596, 300 tonnes of total nitrogen, and 119, 700 tonnes of total phosphorus in 2017. These figures represent 93. 76%, 51. 30%, 42. 14%, and 56. 46% of the total water pollution from agricultural sources, respectively [47]. In addition, projections indicate that livestock carbon emissions constituted 33. 46% of total agricultural carbon emissions in 2019 [47]. Therefore, the animal industry remains a significant contributor to agricultural surface pollution in China. Expediting the green transformation of this industry is essential for its high-quality advancement. For instance, resource recovery from manure is achievable through biogasification and other treatment methods, reducing pollution risks to soil and water bodies. Promoting ecological farming models such as grassland grazing and circular agriculture is also vital to protect natural ecosystems and minimise ecological damage. Moreover, the nation must prioritise the adoption of environmental protection technologies and equipment. This includes energy-saving emission reduction equipment, sewage treatment facilities, and the utilisation of clean energy to reduce the environmental effects of animal production and achieve greener practices. This paper employs five indicators to assess the environmentally friendly development level of China's animal industry: the comprehensive utilisation rate of livestock and poultry manure, the level of household biogas utilisation, total carbon emissions, the carrying capacity of land for livestock and poultry farming, and green total factor productivity [17,48].

Fifth, the level of scientific and technological&management. Advances in information technology have positioned scientific and technological innovation as a prerequisite for enhancing productivity and cultivating high-quality development in the animal industry. For instance, integrating technologies such as the Internet of Things, big data analytics, and artificial intelligence allows for intelligent monitoring and management of livestock production processes. The accuracy feeding technology enables the accurate deployment of feed formulas, optimising feed utilisation. In addition, environmental monitoring technology facilitates real-time tracking of temperature and humidity levels in livestock barns, permitting timely adjustments to maintain

optimal conditions. From the perspective of farming patterns, the traditional family-based small-holder production model is incompatible with the high-quality development in China's animal industry. Eliminating outdated production capacity and expediting the transition towards large-scale livestock and poultry farming are essential steps. Such a shift facilitates the adoption of advanced production technologies and management models, thereby enhancing production efficiency. Moreover, it cultivates economies of scale, reduces production costs, and thus enhances economic efficiency. It is important to recognise that effective management is contingent upon human capital. A highly skilled workforce with a strong environmental awareness and learning capacity is more adept at managing livestock farms. This study employs indicators such as the intensity of mechanical equipment, the scale of livestock and poultry farming, labour force quality, the proportion of senior workers in animal husbandry stations, and the proportion of breeding livestock and poultry stock to comprehensively assess the level of scientific and technological &management in livestock and poultry farming [13,17]. Fig 1 shows the relationship between the indicators for evaluating high-quality development of the animal husbandry.

The data in this paper are mainly from official statistical yearbooks, including China Statistical Yearbook, China Rural Statistical Yearbook, China Animal Husbandry and Veterinary Yearbook, China Environmental Statistical Yearbook, as well as statistical yearbooks and animal husbandry and veterinary yearbooks of each province. According to the division criteria of the National Bureau of Statistics, the eastern region includes Beijing, Tianjin, Hebei, Liaoning, Shanghai, Jiangsu, Zhejiang, Fujian, Shandong, Guangdong, and Hainan;the central region includes Shanxi, Jilin, Heilongjiang, Anhui, Jiangxi, Henan, Hubei, and Hunan; and the western region includes Inner Mongolia, Guangxi, Chongqing, Sichuan, Guizhou, Yunnan, Shanxi, Gansu, Qinghai, Ningxia, and Xinjiang.

## 3. Results

### 3. 1. Descriptive statistics on the high-quality development of China's animal husbandry industry

The overall standard of the high-quality development of animal husbandry in China's provinces remained between 0. 09 and 0. 35 throughout the period from 2010 to 2022(Table 2). Across the 30 provinces, the average level of high-quality development in the animal industry

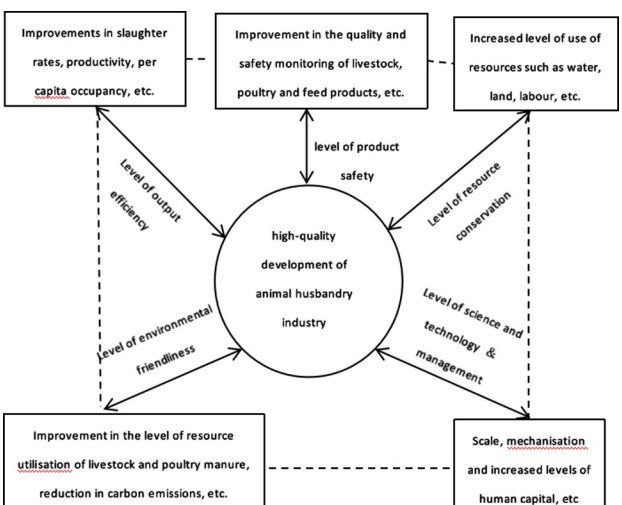

**Fig 1. Relationship of evaluation indicators for high-quality development of the animal industry.**

**Table 2. Level of high-quality development of animal husbandry industry in 30 provinces in China, 2010–2022.**

| | 2010 | Rank | 2012 | Rank | 2014 | Rank | 2016 | Rank | 2018 | Rank | 2020 | Rank | 2022 | Rank | Annual Growth Rate (%) |
|---|---|---|---|---|---|---|---|---|---|---|---|---|---|---|---|
| Anhui | 0. 15 | 8 | 0. 16 | 6 | 0. 16 | 10 | 0. 2 | 5 | 0. 21 | 4 | 0. 25 | 3 | 0. 21 | 5 | 2. 84 |
| Beijing | 0. 22 | 2 | 0. 22 | 2 | 0. 23 | 3 | 0. 23 | 3 | 0. 23 | 3 | 0. 22 | 6 | 0. 25 | 1 | 1. 07 |
| Fujian | 0. 13 | 14 | 0. 15 | 9 | 0. 17 | 6 | 0. 2 | 4 | 0. 21 | 6 | 0. 22 | 5 | 0. 19 | 12 | 3. 21 |
| Gansu | 0. 12 | 19 | 0. 15 | 7 | 0. 16 | 9 | 0. 17 | 10 | 0. 19 | 8 | 0. 19 | 10 | 0. 2 | 9 | 4. 35 |
| Guangdong | 0. 15 | 7 | 0. 15 | 10 | 0. 16 | 11 | 0. 17 | 11 | 0. 17 | 13 | 0. 18 | 13 | 0. 21 | 8 | 2. 84 |
| Guangxi | 0. 13 | 12 | 0. 15 | 11 | 0. 16 | 12 | 0. 16 | 13 | 0. 15 | 18 | 0. 18 | 14 | 0. 19 | 11 | 3. 21 |
| Guizhou | 0. 13 | 13 | 0. 14 | 16 | 0. 14 | 16 | 0. 16 | 15 | 0. 16 | 16 | 0. 16 | 18 | 0. 16 | 19 | 1. 75 |
| Hainan | 0. 14 | 9 | 0. 14 | 13 | 0. 14 | 14 | 0. 17 | 9 | 0. 18 | 9 | 0. 19 | 11 | 0. 22 | 3 | 3. 84 |
| Hebei | 0. 11 | 20 | 0. 11 | 25 | 0. 1 | 28 | 0. 11 | 28 | 0. 11 | 29 | 0. 13 | 27 | 0. 14 | 23 | 2. 03 |
| Henan | 0. 09 | 29 | 0. 1 | 27 | 0. 18 | 5 | 0. 19 | 7 | 0. 18 | 10 | 0. 22 | 7 | 0. 16 | 18 | 4. 91 |
| Heilongjiang | 0. 11 | 22 | 0. 13 | 18 | 0. 13 | 20 | 0. 15 | 17 | 0. 16 | 15 | 0. 18 | 12 | 0. 19 | 10 | 4. 66 |
| Hubei | 0. 12 | 15 | 0. 14 | 12 | 0. 14 | 17 | 0. 16 | 12 | 0. 16 | 14 | 0. 18 | 15 | 0. 18 | 14 | 3. 44 |
| Hunan | 0. 1 | 23 | 0. 11 | 24 | 0. 11 | 25 | 0. 13 | 24 | 0. 13 | 22 | 0. 15 | 22 | 0. 15 | 20 | 3. 44 |
| Jilin | 0. 1 | 27 | 0. 09 | 29 | 0. 09 | 29 | 0. 11 | 27 | 0. 11 | 28 | 0. 13 | 24 | 0. 13 | 26 | 2. 21 |
| Jiangsu | 0. 14 | 10 | 0. 14 | 14 | 0. 26 | 2 | 0. 35 | 2 | 0. 28 | 2 | 0. 33 | 2 | 0. 21 | 6 | 3. 44 |
| Jiangxi | 0. 1 | 24 | 0. 1 | 28 | 0. 11 | 23 | 0. 12 | 25 | 0. 12 | 25 | 0. 15 | 23 | 0. 13 | 25 | 2. 21 |
| Liaoning | 0. 1 | 26 | 0. 11 | 23 | 0. 13 | 21 | 0. 13 | 20 | 0. 12 | 24 | 0. 13 | 26 | 0. 14 | 24 | 2. 84 |
| Inner Mongolia | 0. 16 | 5 | 0. 16 | 5 | 0. 15 | 13 | 0. 16 | 14 | 0. 16 | 17 | 0. 16 | 17 | 0. 18 | 13 | 0. 99 |
| Ningxia | 0. 13 | 11 | 0. 15 | 8 | 0. 14 | 15 | 0. 15 | 16 | 0. 17 | 12 | 0. 19 | 9 | 0. 21 | 7 | 4. 08 |
| Qinghai | 0. 12 | 17 | 0. 13 | 17 | 0. 16 | 8 | 0. 14 | 19 | 0. 18 | 11 | 0. 17 | 16 | 0. 17 | 16 | 2. 95 |
| Shandong | 0. 11 | 21 | 0. 13 | 20 | 0. 11 | 24 | 0. 14 | 18 | 0. 14 | 20 | 0. 16 | 19 | 0. 15 | 22 | 2. 62 |
| Shanxi | 0. 07 | 30 | 0. 09 | 30 | 0. 08 | 30 | 0. 08 | 30 | 0. 1 | 30 | 0. 12 | 29 | 0. 13 | 28 | 5. 29 |
| Shanxi | 0. 1 | 25 | 0. 11 | 22 | 0. 11 | 26 | 0. 12 | 26 | 0. 11 | 26 | 0. 12 | 28 | 0. 12 | 29 | 1. 53 |
| Shanghai | 0. 24 | 1 | 0. 27 | 1 | 0. 37 | 1 | 0. 35 | 1 | 0. 31 | 1 | 0. 35 | 1 | 0. 16 | 17 | -3. 32 |
| Sichuan | 0. 12 | 16 | 0. 14 | 15 | 0. 13 | 19 | 0. 13 | 21 | 0. 15 | 19 | 0. 15 | 20 | 0. 15 | 21 | 1. 88 |
| Tianjin | 0. 15 | 6 | 0. 13 | 19 | 0. 13 | 18 | 0. 13 | 22 | 0. 13 | 23 | 0. 13 | 25 | 0. 12 | 30 | -1. 84 |
| Xinjiang | 0. 09 | 28 | 0. 1 | 26 | 0. 1 | 27 | 0. 1 | 29 | 0. 11 | 27 | 0. 12 | 30 | 0. 13 | 27 | 3. 11 |
| Yunnan | 0. 16 | 4 | 0. 16 | 4 | 0. 17 | 7 | 0. 18 | 8 | 0. 2 | 7 | 0. 22 | 8 | 0. 22 | 4 | 2. 69 |
| Zhejiang | 0. 12 | 18 | 0. 12 | 21 | 0. 12 | 22 | 0. 13 | 23 | 0. 14 | 21 | 0. 15 | 21 | 0. 17 | 15 | 2. 95 |
| Chongqing | 0. 16 | 3 | 0. 17 | 3 | 0. 19 | 4 | 0. 19 | 6 | 0. 21 | 5 | 0. 24 | 4 | 0. 22 | 2 | 2. 69 |

has exhibited an upward trend, rising from 0. 13 in 2010 to 0. 17 in 2022(Fig 2). Analysing specific dimensions(see Fig 3), the level of scientific and technological &management in China's livestock and poultry farming demonstrates a clear upward trend, achieving the highest level across all five dimensions. The level of resource conservation has remained relatively consistent, maintaining a relatively high level across the five dimensions. Environmental performance demonstrates a clear pattern of improvement, reflecting the increasing effectiveness of environmental management strategies in the animal huabandry industry and the positive effect of sustainable livestock farming practices. The level of output efficiency has remained stable but at a lower level, suggesting that there is significant potential for improvement in the productivity of China's animal huabandry industry. The safety of livestock products, while currently at a relatively low level, exhibits a clear upward trend. This suggests that the increasingly rigorous oversight of animal production in China, coupled with stringent measures against the prohibited drug use, and the overuse of antibiotics, has yielded improvements in the safety of China's animal products. However, some quality and safety concerns persist, leading to varying degrees of risk to public health from meat, eggs, and milk.

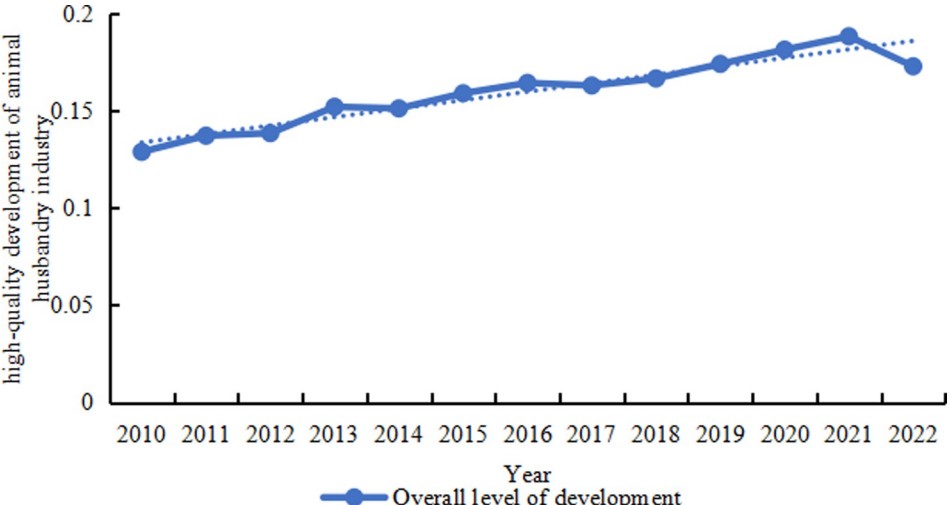

**Fig 2. Provincial averages of high quality levels of animal husbandry industry in China, 2010–2022.**

In addition, this paper employs a systematic clustering methodology (Q-type cluster analysis) to categorise the level of high-quality development in the animal husbandry industry across provinces [49,50]. Based on the systematic clustering analysis results and considering the average level of high-quality development in the animal husbandry industry of each province from 2010 to 2022, the 30 provinces are grouped into four categories: high-quality animal husbandry industry areas, medium-quality animal husbandry industry areas, medium- low quality animal husbandry industry areas, and low-quality animal husbandry industry areas (see Table 3 for results).

Type I:High-quality animal husbandry industry areas. The average level of high-quality development of the animal husbandry industry in 2010–2022 is between 0. 25 and 0. 31, with only two regions, Jiangsu and Shanghai, included in this category. As provinces renowned for their economic strength, advanced technologies, and robust educational systems in China,

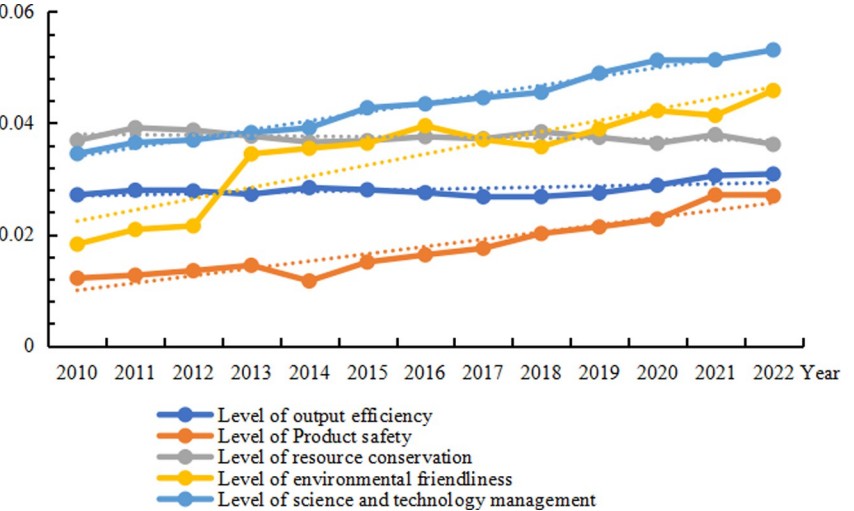

**Fig 3. Provincial averages of five dimensions of high-quality development of China's animal husbandry industry, 2010–2022.**

**Table 3. Classification of 30 provinces in China into categories of high quality development level of animal husbandry industry, 2010–2022.**

| categories | provinces |
|---|---|
| high-quality areas of the animal husbandry industry(0. 25–0. 31) | Jiangsu, Shanghai |
| medium-quality areas of the animal husbandry industry(0. 18–0. 24) | Anhui, Beijing, Fujian, Yunnan, Chongqing |
| medium-low quality level of animal husbandry industry(0. 14–0. 18) | Gansu, Guangdong, Guangxi, Guizhou, Hainan, Henan, Honglongjiang, Hubei, Inner Mongolia, Ningxia, Qinghai |
| low-quality areas of livestock husbandry industry(0. 09–0. 14) | Hebei, Hunan, Jilin, Jiangxi, Liaoning, Shandong, Shanxi, Shanxi, Sichuan, Tianjin, Xinjiang, Zhejiang |

their progress in livestock farming can be primarily attributed to a superior environmental friendliness index, comprehensive management of livestock and poultry waste, and higher green total factor productivity. Shanghai, an international metropolis with leading scientific research institutions and a skilled workforce, prioritises environmental protection and has enacted various environmental management measures [51]. These include rigorous measures to reduce air and water pollution alongside active ecological restoration and greening efforts, finally cultivating a greener and more sustainable animal husbandry industry. Jiangsu Province, characterised by its diverse animal husbandry models, centres its approach on scientific and technological advancements, industry upgrades, ecological farming practices, standardised management, and brand development. In addition, the province has implemented a suite of policies promoting animal husbandry industry expansion, including financial subsidies, tax incentives, and scientific and technological support [52], offering valuable insights for the advancement of animal husbandry in other provinces.

The type II:Medium-quality animal husbandry industry areas. Exhibiting a high-quality development level in the 0. 18–0. 24 range between 2010 and 2022, this group comprises five provinces: Anhui, Beijing, Fujian, Yunnan, and Chongqing. The moderately high ranking of these five provinces is primarily attributed to their advanced technological and managerial expertise in animal husbandry. While not considered major hubs for animal production in China, these provinces demonstrate a relatively high level of high-quality development in their respective animal husbandry industries. Anhui and Beijing stand out for the considerable scale of their livestock operations. Fujian exhibits a significant scale level and maintains a significant proportion of breeding livestock and poultry stock nationally. Yunnan and Chongqing are recognised for their higher levels of mechanisation. Beijing benefits from robust capital, advanced technology, science, and a skilled workforce, resulting in a sophisticated level of animal husbandry development. Pig and poultry farming constitute the backbone of Anhui province's animal husbandry. Influenced by environmental regulations, smaller farms are progressively phased out, paving the path for the expansion of large-scale, modern farming practices [53,54], thereby elevating the overall development of animal husbandry in the province. Fujian Province's animal husbandry mainly revolves around poultry production, with a particular emphasis on chicken rearing [55]. This specialization contributes to a lower developmental threshold for the industry. Yunnan Province has a diverse range of livestock species and abundant pastureland, presenting favourable conditions for the advancement of animal husbandry [56]. In recent years, the Chongqing Municipal Government has implemented a series of supportive policies to cultivate the growth of its animal husbandry industry. Simultaneously, higher requirements for environmental protection and animal welfare have contributed to the steady improvement of high-quality development in the industry [57].

The type III:Medium- low quality animal industry areas. This group, with a high-quality development index ranging from 0. 14 to 0. 18 between 2010 and 2022, includes eleven provinces:Gansu, Guangdong, Guangxi, Guizhou, Hainan, Henan, Heilongjiang, Hubei, Inner Mongolia, Ningxia, and Qinghai. It is worth noting that Heilong jiang, Inner Mongolia, Hubei, and Henan, while being major animal-producing provinces in China, demonstrate a relatively low level of high-quality development in their animal husbandry industries. This can be attributed to several constraining factors, including underdeveloped animal husbandry infrastructure, cold climatic conditions, a lack of environmental awareness, and a deficiency in high-quality personnel in Heilongjiang. Similarly, Ningxia and Gansu face limitations in achieving a high level of high-quality development in animal husbandry due to factors such as the relative scarcity of water resources [58]. The average annual growth rate of the high- quality development level in the significant pastoral areas of Inner Mongolia and Qinghai remains low. This slower rate of development reflects the challenges faced in these regions, including issues such as overgrazing, the irrational utilisation of pastureland, arid climatic conditions, and the relative scarcity of water resources [59–61]. Besides, the persistence of traditional nomadic pastoralism practices in certain areas, combined with a lack of modern breeding methods and management approaches, negatively affects the overall development and effectiveness of the animal husbandry sector in Inner Mongolia and Qinghai regions [62].

The type IV:Low-quality animal husbandry industry areas. Between 2010 and 2022, these regions, spanning twelve provinces including Hebei, Hunan, Jilin, Jiangxi, Liaoning, Shandong, Shanxi, Shanxi, Sichuan, Tianjin, Xinjiang and Zhejiang, exhibited a high-quality development index ranging from 0. 09 to 0. 14. Specifically, key livestock farming provinces such as Hunan, Shandong, Sichuan, Xinjiang, Hebei, Zhejiang, Liaoning, and Jilin, registered the lowest levels of high-quality development in the national animal husbandry industry. This points to a prevalent issue in Chinese animal husbandry industry:a tendency towards"big production but not good quality". It is not difficult to find that the southern water network region(The southern water network area mainly involves 10 provinces, including Shanghai, Jiangsu, Zhejiang, Anhui, Jiangxi, Shandong, Henan, Hubei, Hunan and Guangdong.) as an important strategic area of China's pig farming, in addition to Shanghai, Jiangsu and Anhui's higher level of animal husbandry development, the level of animal husbandry development in other regions are generally low, reflecting that pressure for high-quality transformation of the animal husbandry sector remains high in the water network region of southern China. Considering the significant differences in fundamental conditions for animal husbandry development across these provinces, customized approaches to cultivating high-quality advancement are essential. For instance, Shandong Province, a major beef cattle breeding region characterised by extensive farming practices and a high concentration of farms, suffers primarily from environmental pollution. Therefore, future efforts should prioritise the sustainable development of animal husbandry in the region. This can be achieved through a greater emphasis on manure resource utilisation and recycling technologies, alongside the promotion of environmentally sound feed options;whereas, Hunan and Zhejiang, as major pig farming provinces, should focus on optimizing the regional distribution of pig farming operations. This involves enhancing standardised large-scale farming practices and encouraging the establishment of integrated crop-livestock ecosystems that prioritise ecological recycling principles.

## 3. 2. Regional differences in high-quality development of China's animal husbandry industry and their sources

To gain a deeper understanding of the regional differences in the level of high- quality development in China's animal husbandry industry, this study evaluates the overall differences

observed between 2010 and 2022. Employing Dagum's Gini coefficient as the primary analytical tool, the research evaluates inter-regional differences, intra-regional differences, hypervariable density difference and the specific contributions of differences in inter-region differences, intra-region differences, hypervariable density to the overall differences identified. The comprehensive findings derived from these computations are systematically presented in Table 4.

Fig 4 illustrates both the overall differences in the high-quality development of China's animal husbandry industry and its development. The data indicates a pattern of "slow decline-rise-slow decline" in overall regional differences during the observation period. Utilizing 2010 as a baseline, regional difference contracted by 1. 28% annually. This overall difference steadily decreased from 0. 14 in 2010 to 0. 13 in 2012, before a slight increase to 0. 18 in 2017. Following this, a consistent decline brought it to a nadir of 0. 12 in 2022. Broadly, regional differences in the high-quality development of China's animal husbandry industry have remained relatively minor in recent years, fluctuating between 0. 12 and 0. 18 and exhibiting a marginally downward trend.

Fig 5 illustrates the intra-regional differences and evolutionary trends of high-quality development in the animal husbandry industry across the three primary regions of East, Central, and West. Throughout the observed timeframe (2010–2022), the East exhibited the most significant intra-regional differences, followed by the Central and then the West. Regarding developmental trends, a degree of similarity exists across the three regions. The East demonstrates a pattern of "stable—rapid rise—fluctuating stable—slow decline. " Intra-regional discrepancies remained at 0. 15 from 2010 to 2012, sharply increased to 0. 21 between 2012 and 2013, oscillated between 0. 19 and 0. 22 from 2014 to 2021, and finally decreased to 0. 13 in 2022. Specifically, Shanghai, Jiangsu, Beijing, and Fujian stand out as high to medium-quality development zones in the animal husbandry industry;whereas, Guangdong and Hainan represent areas of medium-low quality, while Hebei, Liaoning, Shandong, Zhejiang, and Tianjin constitute low-quality development areas. This is related to the significant differences among eastern provinces in geographical location, resource availability, policy support, industrial development strategies, and consumer demand. The Central region's development trend reflects this pattern with a"stable-slowly rising- slowly falling. " Intra-regional differences in this region held steady at 0. 11 from 2010 to 2012, climbed from 0. 11 to 0. 16 between 2012 and 2017, and fell from 0. 16 to 0. 10 between 2017 and 2022. Anhui appears to the leading

**Table 4. Regional differences in high-quality development of China's animal husbandry industry and sources of differences, 2010–2022.**

| Year | Intra-region | | | Inter-region | | | contribution rate(%) | | |
|------|-------------------|--------|--------|--------|--------------|--------------|--------------|--------------|--------------|--------------|
| | overall differences | eastern | central | western | Eastern-central | Eastern-western | Central-western | Intra-regional | Inter-regional | hypervariable |
| 2010 | 0. 14 | 0. 15 | 0. 11 | 0. 10 | 0. 19 | 0. 14 | 0. 14 | 29. 95 | 47. 07 | 22. 97 |
| 2011 | 0. 13 | 0. 15 | 0. 11 | 0. 09 | 0. 18 | 0. 13 | 0. 13 | 30. 11 | 47. 88 | 22. 01 |
| 2012 | 0. 13 | 0. 15 | 0. 11 | 0. 08 | 0. 17 | 0. 13 | 0. 13 | 30. 03 | 38. 85 | 31. 12 |
| 2013 | 0. 16 | 0. 21 | 0. 13 | 0. 08 | 0. 21 | 0. 18 | 0. 12 | 31. 59 | 40. 22 | 28. 19 |
| 2014 | 0. 17 | 0. 22 | 0. 13 | 0. 09 | 0. 22 | 0. 19 | 0. 13 | 31. 45 | 41. 46 | 27. 09 |
| 2015 | 0. 17 | 0. 21 | 0. 15 | 0. 09 | 0. 22 | 0. 18 | 0. 13 | 31. 31 | 37. 91 | 30. 78 |
| 2016 | 0. 18 | 0. 22 | 0. 15 | 0. 10 | 0. 22 | 0. 20 | 0. 13 | 31. 55 | 38. 32 | 30. 14 |
| 2017 | 0. 18 | 0. 22 | 0. 16 | 0. 10 | 0. 22 | 0. 18 | 0. 14 | 31. 78 | 35. 39 | 32. 84 |
| 2018 | 0. 16 | 0. 19 | 0. 14 | 0. 10 | 0. 19 | 0. 16 | 0. 13 | 32. 11 | 29. 31 | 38. 58 |
| 2019 | 0. 16 | 0. 20 | 0. 14 | 0. 11 | 0. 19 | 0. 18 | 0. 13 | 32. 36 | 26. 46 | 41. 18 |
| 2020 | 0. 16 | 0. 20 | 0. 13 | 0. 11 | 0. 18 | 0. 17 | 0. 13 | 33. 02 | 21. 18 | 45. 80 |
| 2021 | 0. 15 | 0. 19 | 0. 12 | 0. 10 | 0. 17 | 0. 17 | 0. 12 | 32. 48 | 22. 02 | 45. 50 |
| 2022 | 0. 12 | 0. 13 | 0. 10 | 0. 10 | 0. 13 | 0. 12 | 0. 11 | 33. 09 | 17. 23 | 49. 68 |

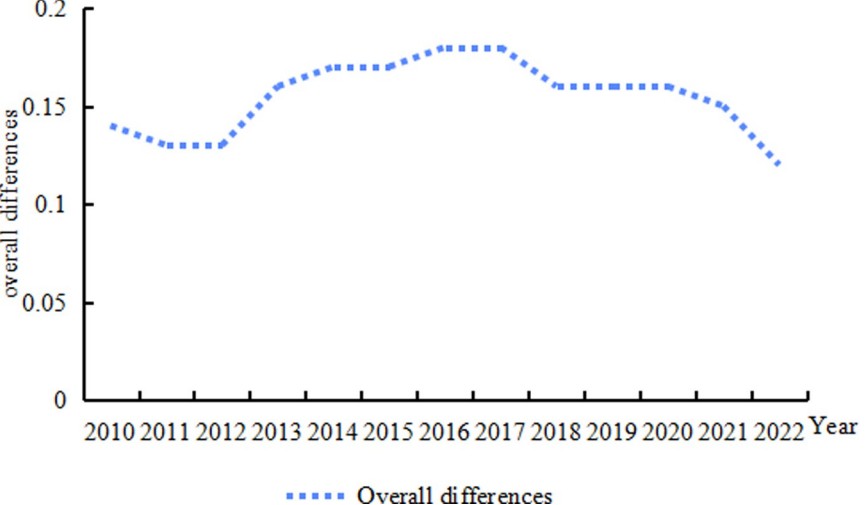

**Fig 4. Trends in the evolution of overall differences in the level of high-quality development of China's animal husbandry industry, 2010–2022.**

region for animal husbandry development in the Central region. In contrast, Henan, Heilongjiang, and Hubei represent medium-low quality development areas, while Hunan, Jilin, Jiangxi, and Shanxi constitute low-quality zones. Finally, the West demonstrates relative stability in its intra-regional differences, hovering between 0. 08 and 0. 11. Yunnan and Chongqing have the highest development levels, placing them in the medium-quality tier. Gansu, Guangxi, Guizhou, Inner Mongolia, Ningxia, and Qinghai occupy the medium-low quality tier, whereas Shaanxi, Sichuan, and Xinjiang represent low-quality development areas in the animal husbandry sector.

Fig 6 further illustrates the inter-regional differences and evolutionary trends in the high-quality development of China's animal husbandry industry. The most significant inter-regional differences are observed between the eastern and central regions, exhibiting a pattern

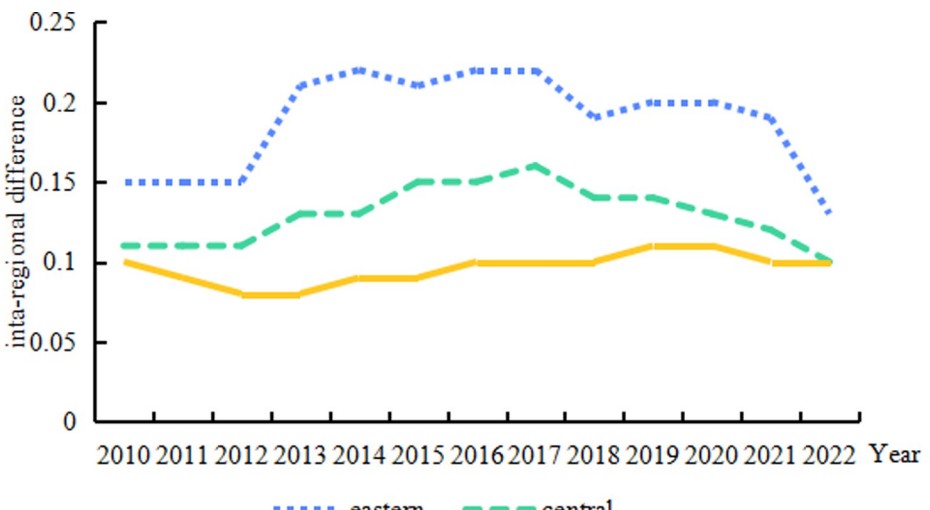

**Fig 5. Trends in the evolution of intra-regional differences in the level of high-quality development of China's animal husbandry industry, 2010–2022.**

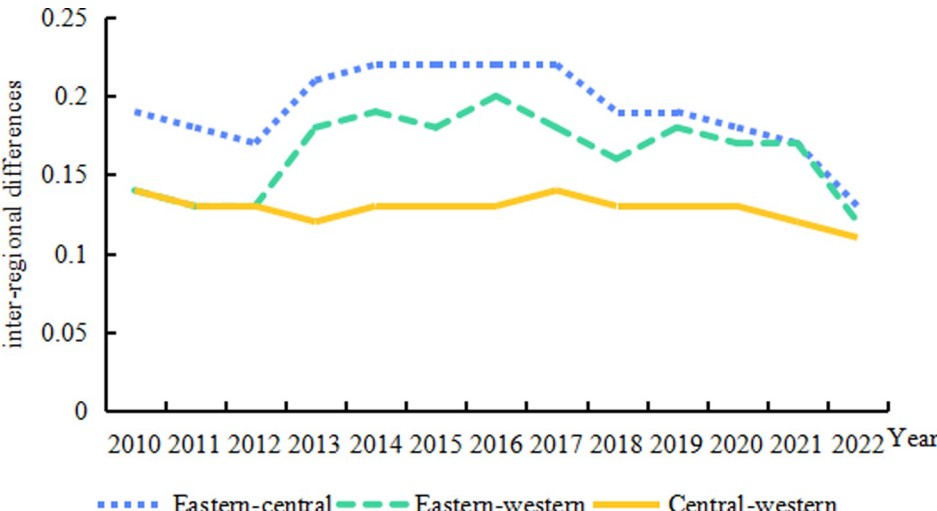

**Fig 6. Trends in the evolution of inter-regional differences in the level of high-quality development of China's animal husbandry industry, 2010–2022.**

of"falling-fluctuating-rising-steady-fluctuating"falling. This pattern is characteris ed by a decrease from 0. 19 in 2010 to 0. 17 in 2012, followed by an increase to 0. 22 in 2014. A period of stability ensues from 2014 to 2017, succeeded by a fluctuating decline to 0. 13 between 2017 and 2022. Similarly, the eastern and western regions demonstrate a"fluctuating upward-steady-fluctuating downward"trend. This trend comprises a decline from 0. 14 in 2010 to 0. 13 in 2012, followed by a fluctuating increase to 0. 20 in 2016, and a decline to 0. 12 in 2022. The smallest difference is identified in the regional difference between the central and western regions, presenting a relatively stable, albeit slightly reducing, trend from 0. 14 in 2010 to 0. 11 in 2022. At a macro level, the relatively small gap in animal husbandry industry development between the central and western regions can be attributed to the similar levels of natural resource endowment and socio-economic development in these regions;whereas, the eastern region's significantly advanced overall level of social development, coupled with its unique advantages in animal husbandry, results in its high-quality animal husbandry development being significantly ahead of the central and western regions.

Fig 7 presents the specific contributions of differences in inter-region differences, intra-region differences, hypervariable density to the overall differences identified. The contribution of intra-regional differences to the total variance demonstrates a minor upward trend, from 29. 95% in 2010 to 33. 09% in 2022;whereas, the contribution rate of inter-regional differences to the total difference exhibits a reducing trend, declining from 47. 07% in 2010 to 17. 23% in 2022, averaging an annual decrease of 8. 03%. This signifies that inter-regional differences exert a waning influence on the overall difference, with 2022 marking their least effect. In contrast, the contribution of hypervariable density differences to the total difference is on the rise, increasing from 22. 97% in 2010 to 49. 68% in 2022. Overall, the source of regional difference in the high-quality development of the animal husbandry industry has significantly changed, transforming from inter-region > intra-region > hypervariable density in 2010 to hypervariable density > intra-region > inter-region in 2022. The degree of influence exerted by intra-regional differences on the overall differences remains relatively stable. The effect of inter-regional differences on the overall differences gradually leans towards being the least impactful. The degree of influence of hypervariable density on the overall differences gradually tends towards being the most impactful. Specifically, whether in the eastern, central, or western

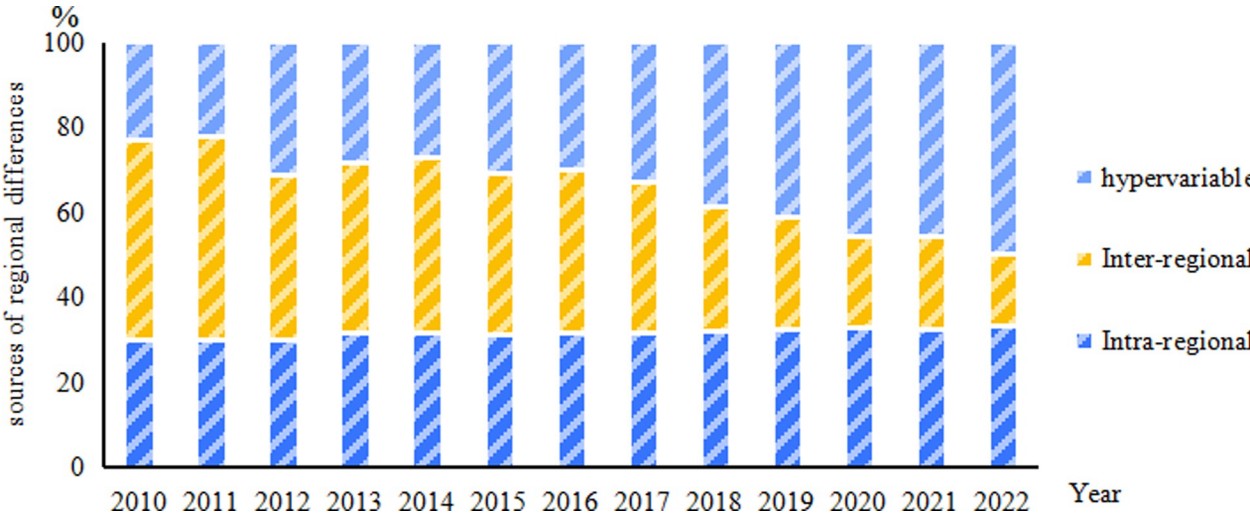

**Fig 7. Sources of regional differences in the level of high-quality development of China's animal husbandry industry, 2010–2022.**

regions, each region comprises provinces exhibiting both relatively high and low levels of high-quality development in animal husbandry industry [63,64].

### 3. 3. Characteristics of the temporal dynamic evolution of high-quality development of China's animal husbandry industry

This paper utilises Stata software 17. 0 to map the kernel density of high-quality development in the animal husbandry industry across 30 Chinese provinces over the period 2010 to 2022. The kernel density map of the level of high-quality development of the national animal husbandry industry is characterised by two features. Firstly, the peak of the kernel density function demonstrates a progressive shift towards the right throughout the observed timeframe. In 2012, the high-quality development level of the industry across the 30 provinces was largely concentrated in the 0–0. 28 range. However, by 2022, this concentration had shifted to the 0. 10–0. 40 range. This shift signifies a gradual enhancement in the overall high-quality development level of the animal husbandry industry nationwide. Secondly, the peak of the kernel density curve exhibits a general downward trend, with a twin-peaked curve trend in 2022. This pattern suggests a rising equilibrium in the high-quality development level of China's animal husbandry industry, accompanied by a certain gradient effect(see Fig 8).

The distribution dynamics associated with high-quality development in the eastern region's animal husbandry industry exhibit two main characteristics. Firstly, the kernel density curve presents with a prominent primary peak and a secondary peak in 2012. The considerable altitude of the primary peak signifies a significant gradient effect, highlighting imbalance in the level of high-quality development across the eastern region's livestock industry in 2012. Secondly, the peak of the kernel density graph in 2012–2022 shows a decreasing and then an increasing trend, while the overarching trend remains downwards. This suggests a narrowing of the differences in high-quality development levels in the eastern region's animal husbandry industry throughout this period(see Fig 9). The distribution range of the kernel density map of the high-quality development level of the animal husbandry industry in the central region is relatively narrow. It shows a rightward trend, with a distribution range of 0. 07–0. 16 in 2012 and a distribution range of 0. 17–0. 22 in 2022. The height of the peaks shows a"downward-ascending" trend. This reflects the fact the distribution range of the high-quality development

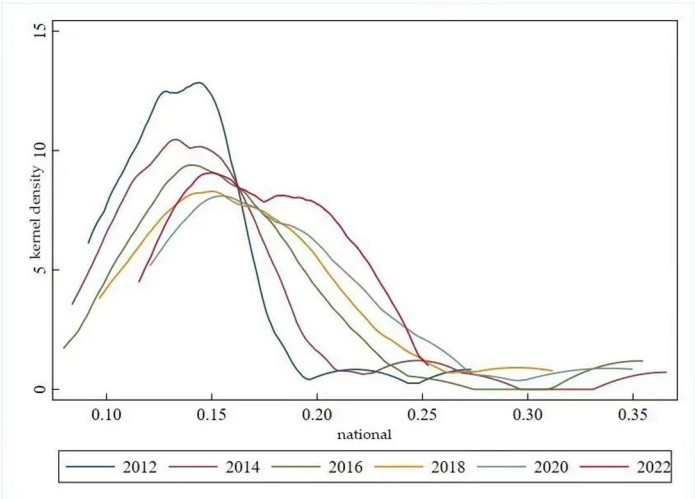

**Fig 8. Kernel density map for high-quality development.**

level of the animal husbandry industry in the central region is also more concentrated, showing a clear positive trend, and the level of balanced distribution has slightly increased(see Fig 10). The kernel density plot charting high-quality development in the western region's animal husbandry industry indicates downward trend in peak height accompanied by a rightward shift in distribution range. This pattern suggests a tendency towards an increased level of high-quality development in the western region's animal husbandry industry, accompanied by an improvement in the overall equilibrium(see Fig 11).

According to the comprehensive assessment of animal husbandry industry practices, a four-tier classification system has been established to categorise the level of development across 30 provinces in China. Provinces achieving a comprehensive measurement score in the interval of [0, 0. 1258] are designated as Tier 1, indicating a low level of animal husbandry induetry development. A score in the interval of (0. 1258, 0. 1510] corresponds to Tier 2,

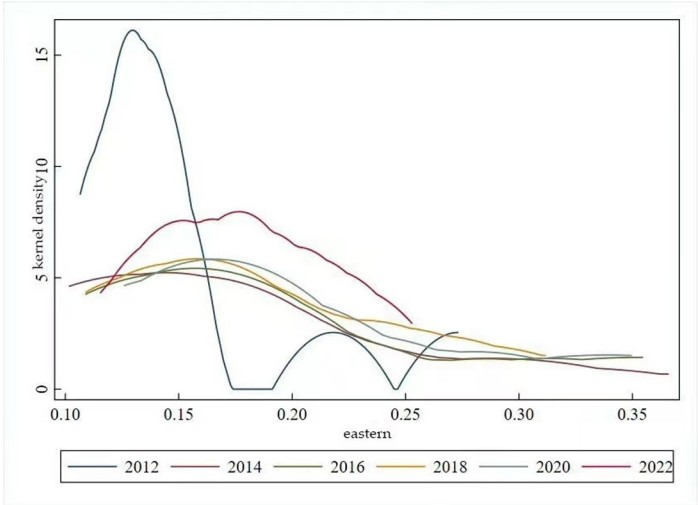

**Fig 9. Kernel density map for high-quality development of the national livestock husbandry of the eastern livestock husbandry.**

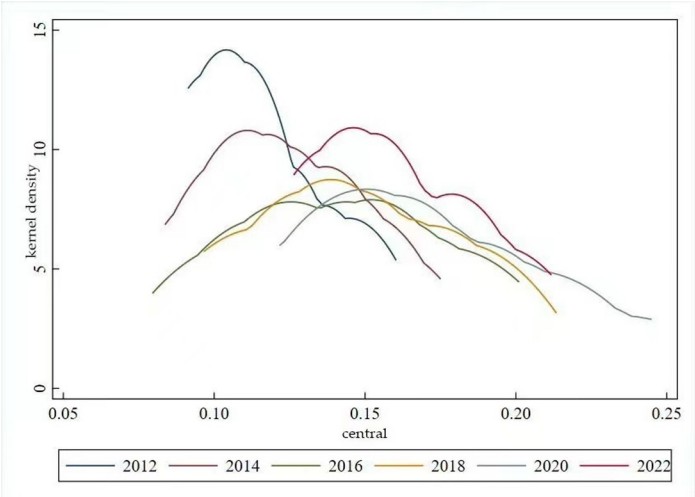

**Fig 10. Kernel density map for high-quality development.**

representing a medium-low level of development. Provinces attaining a score in the interval of (0. 1510, 0. 1808] are classified as Tier 3, signifying a medium level of development. Finally, provinces achieving a score in the interval of (0. 1808, +∞) are grouped under Tier 4, indicating a high level of animal husbandry development.

As can be seen from Table 5, the dynamic evolution of the high-quality development level of the animal husbandry industry without considering geospatial factors is characterised as follows:Firstly, the internal mobility among the distribution of high-quality development levels in the animal husbandry industry is limited. The values located diagonally in the low, medium-low, medium, and high-level groups (0. 750, 0. 642, 0. 724, and0. 971, respectivel y)are significantly larger than their non-diagonal counterparts. This suggests a significant "path dependence" and "self-locking" effect [65,66]. in the high-quality development levels of China's provincial animal husbandry industries. Secondly, the transformation among levels of high-

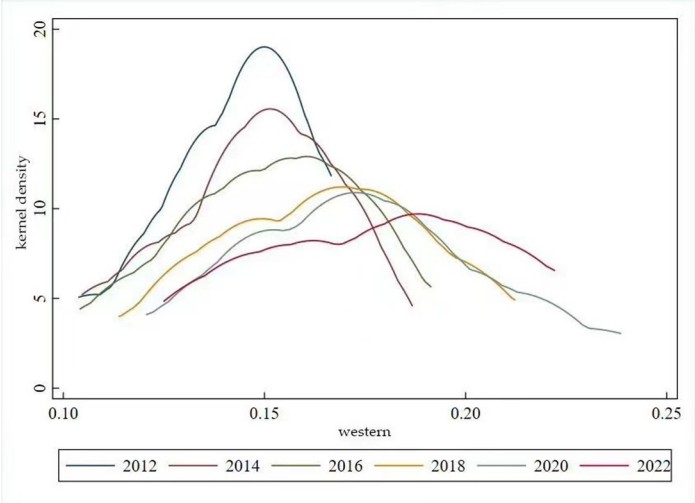

**Fig 11. Kernel density map for high-quality development of the central livestock husbandry of the western livestock husbandry.**

**Table 5. Markov chain transfer distribution for high-quality development of China's animal husbandry industry.**

| Neighbourhood type | | low level | medium-low level | medium level | high level | n |
|---|---|---|---|---|---|---|
| Traditional | low level | 0. 750 | 0. 250 | 0. 000 | 0. 000 | 84 |
| | medium-low level | 0. 025 | 0. 642 | 0. 333 | 0. 000 | 81 |
| | medium level | 0. 013 | 0. 053 | 0. 724 | 0. 211 | 76 |
| | high level | 0. 000 | 0. 000 | 0. 029 | 0. 971 | 69 |
| low level | low level | 0. 894 | 0. 106 | 0. 000 | 0. 000 | 47 |
| | medium-low level | 0. 000 | 0. 750 | 0. 250 | 0. 000 | 12 |
| | medium level | 0. 000 | 0. 000 | 0. 750 | 0. 250 | 4 |
| | high level | 0. 000 | 0. 000 | 0. 000 | 1. 000 | 0 |
| medium-low level | low level | 0. 667 | 0. 333 | 0. 000 | 0. 000 | 27 |
| | medium-low level | 0. 000 | 0. 808 | 0. 192 | 0. 000 | 26 |
| | medium level | 0. 000 | 0. 083 | 0. 750 | 0. 167 | 12 |
| | high level | 0. 000 | 0. 000 | 0. 000 | 1. 000 | 11 |
| medium level | low level | 0. 143 | 0. 857 | 0. 000 | 0. 000 | 7 |
| | medium-low level | 0. 063 | 0. 531 | 0. 406 | 0. 000 | 32 |
| | medium level | 0. 031 | 0. 063 | 0. 719 | 0. 188 | 32 |
| | high level | 0. 000 | 0. 000 | 0. 000 | 1. 000 | 20 |
| high level | low level | 0. 667 | 0. 333 | 0. 000 | 0. 000 | 3 |
| | medium-low level | 0. 000 | 0. 455 | 0. 545 | 0. 000 | 11 |
| | medium level | 0. 000 | 0. 036 | 0. 714 | 0. 250 | 28 |
| | high level | 0. 000 | 0. 000 | 0. 053 | 0. 947 | 38 |

quality development in the animal husbandry industry primarily reflects upward shifts between contiguous subgroups, while the probability of trans-group movements remains insignificant. The low-level group, for instance, presents a 0. 250 probability of transitioning to the medium-low level group, with the probability of transitioning to the medium and high-level groups being insignificant. Comparatively, the medium-low level group demonstrates a 0. 333 probability of advancing to the medium level group, while the probability of regressing to the low level group is only 0. 025, and, again, the probability of reaching the high level group is insignificant. Similarly, the medium level group demonstrates a 0. 211 probability of ascending to the high level group, while the probabilities of descending to the medium-low level group (0. 053) and the low-level group (0. 013)are insignificant. Thirdly, there is a certain degree of "Matthew effect" in the dynamic evolution of the level of high-quality development of the animal husbandry industry [67]. The probabilities of the high-level group and low-level group maintaining their current state are 0. 971 and 0. 750 respectively, which is significantly higher than the probabilities of the medium-level group (0. 724) and medium-low level group (0. 642) maintaining their current states. That is to say, it is very easy for the high level group area to keep the leading state, but it is difficult for the low level group area to get rid of the lagging state.

The transfer probability in the spatial Markov chain exhibits more significant differences compared to the traditional Markov chain model. For instance, when the effect of neighbouring areas is not factored in, the probability of a low-level group transitioning to a low-middle level group stands at 0. 250. However, when a low-level group is spatially adjacent to low-level, medium-low level, medium-level, and high-level group areas, the probabilities associated with this upward transition are 0. 106, 0. 333, 0. 857, and 0. 333, respectively. This finding indicates that the probability of an upward transition for a low-level group decreases when bordered by other areas of similar, low-level status; whereas, the potential for upward transition is significantly greater when these low-level groups border medium-low level, medium-level, and high-

level groups, with the probability reaching its peak when adjacent to a medium-level group. When the spatial influence of neighbouring areas is not considered, the upward transition probability for medium-low level groups is calculated to be 0. 333. This probability is significantly changed to 0. 250, 0. 192, 0. 406, and 0. 545 when bordering low- level, medium-low level, medium -level, and high-level groups, respectively. This analytical incorporation of the spatial factor indicates that the proximity to both low and medium-low level groups negatively affect the probability of upward transition for medium-low level groups, causing a decrease. This probability only exhibits an increase when bordering medium and high-level groups, reaching its highest point when adjacent to a high-level group. For areas categorically classified as medium- level, the probability of experiencing an upward transition is 0. 211 when the effect of neighbouring areas is disregarded. This probability changes, becoming 0. 250, 0. 167, 0. 188, and 0. 250 when bordering low-level, medium-low level, medium-level, and high-level groups, respectively. It worth noting that the probability of upward transition for the medium-level group displays only a marginal increase when it neighbours a high-level group. For a high-level group, the probability of remaining stably in the same group is almost 1 when adjacent to low-level, medium-low level, medium-level, or high-level groups. This observed trend reflects the fact that the development of the animal husbandry industry in high-level groups is highly stable and exhibits characteristics of path dependency, with the probability of a downward transition being highly insignificant.

## 3. 4 Characteristics of the spatial correlation of the high-quality development of China's animal husbandry industry

Based on the modified gravity model, this paper calculates the strength of correlation of high-quality development of animal husbandry industry among provinces [68,69], and uses social network analysis to portray the characteristics of the network structure and the evolution trend of China's high-quality development level of animal husbandry industry from 2010–2022. Table 6 shows the spatial network density and correlation of high-quality development of China's animal husbandry industry from 2010 to 2022.

Network density is the ratio between the number of links that actually exist in the network and the theoretical maximum number of links that can exist, which is used to measure the

**Table 6. Characteristics of spatial correlation network for high-quality development of China's animal husbandry industry.**

| Year | number of associations | Density | Correlation analysis | | |
|------|------------------------|---------|-------------|-----------|------------|
| | | | Connectedness | Hierarchy | Efficiency |
| 2010 | 177 | 0. 20 | 1 | 0. 59 | 0. 71 |
| 2011 | 181 | 0. 21 | 1 | 0. 55 | 0. 70 |
| 2012 | 184 | 0. 21 | 1 | 0. 59 | 0. 69 |
| 2013 | 190 | 0. 22 | 1 | 0. 59 | 0. 68 |
| 2014 | 191 | 0. 22 | 1 | 0. 72 | 0. 68 |
| 2015 | 193 | 0. 22 | 1 | 0. 62 | 0. 67 |
| 2016 | 190 | 0. 22 | 1 | 0. 65 | 0. 68 |
| 2017 | 188 | 0. 22 | 1 | 0. 68 | 0. 68 |
| 2018 | 183 | 0. 21 | 1 | 0. 71 | 0. 70 |
| 2019 | 187 | 0. 21 | 1 | 0. 71 | 0. 69 |
| 2020 | 179 | 0. 21 | 1 | 0. 71 | 0. 70 |
| 2021 | 184 | 0. 21 | 1 | 0. 71 | 0. 69 |
| 2022 | 189 | 0. 22 | 1 | 0. 71 | 0. 68 |

closeness and complexity of the network [70]. The number of links between provinces in the high-quality development of animal husbandry network has increased from 177 in 2010 to 189 in 2022. The density of the network remains at a low level between 0. 20 and 0. 22, but shows a slight increasing trend, indicating that the degree of linkage of the high-quality development level of the animal husbandry industry between provinces shows a trend of tightening and complexity, but is still at a low level. The mobility of the relevant production factors and resources in each province is weak, and the degree of spatial association of the high-quality development level of animal husbandry industry between provinces still has a large room for improvement, and the efficient reorganisation and allocation of interregional resource factors is expected to be further strengthened. The connectedness degree [71] has been 1 during the sample period, indicating that there are no isolated provinces in the high-quality development level of China's animal husbandry industry and all provinces have established certain links with other provinces. The degree of network hierarchy [72] has always remained at a high level, rising from 0. 59 in 2010 to 0. 71 in 2022, reflecting the fact that the level of high-quality development of China's animal husbandry industry shows obvious hierarchical characteristics, which are becoming more and more strict. The degree of differentiation in the level of high-quality development of China's animal husbandry industry is also becoming more and more obvious, and the "spatial spillover effect" and "diffusion effect" need to be further strengthened. Network efficiency remains at a high level between 0. 67 and 0. 71 and shows a slight downward trend, reflecting the existence of certain redundant relationships and multiple superimposed spillover channels in the level of high-quality development of China's animal husbandry industry(see Table 6).

Table 7 shows the outdegree, indegree, closeness, betweenness and their rankings for 30 provinces in China in 2010 and 2022. In terms of the outdegree [73], the outdegree of each province remained between 1–9 in 2010 and between 2–10 in 2022, showing a stable trend, reflecting that the spatial spillover effects and diffusion effects of China's high-quality development of animal husbandry industry did not show a trend of intensification in the period from 2010 to 2022. In terms of specific provinces, Fujian, Guizhou, Shanghai, Henan, Hunan, Gansu, Guangdong, Heilongjiang, Ningxia, Qinghai, Sichuan, Xinjiang, Yunnan, Chongqing, Hubei and Guangxi remain above seven provinces, which are among the provinces with higher point outdegree. It is not difficult to find that the provinces with high point outdegree are mainly the main producing provinces of China's animal husbandry industry, even though the level of high- quality development of the anmal husbandry industry in these provinces is not very high. Hainan, Jiangsu, Jiangxi, Jilin, Shandong, Shanxi, Shanxi, and Zhejiang are in a medium level state in terms of the spillover effect of the level of high-quality development of the animal husbandry industry. Jiangsu, which has a good level of high-quality development of animal husbandry instudry, has insufficient spatial radiation-driven capacity for high-quality development of the animal husbandry industry in the surrounding areas. Anhui, Hebei, Liaoning, Inner Mongolia, Tianjin, Beijing's point outdegree is in the 4 or less, the spatial spillover ability to remain in the bottom level state.

In terms of the indegree [73], the average point indegree of each province has increased from 5 in 2010 to 6 in 2022. The point indegree of Shanghai, Beijing, Jiangsu, Zhejiang has always been in the top four, maintained in more than 20 provinces, and the point indegree is significantly larger than the point outdegree, reflecting that the"Siphon effect"of the high- quality development of animal husbandry industry in Shanghai, Beijing, Jiangsu, Zhejiang is larger than the "trickle-down effect". Guangdong and Tianjin, whose point indegree is maintained at around 10 provinces, also have a strong "Siphon effect". Hainan, Heilongjiang, Jilin, Liaoning, Inner Mongolia, Ningxia, Gansu, Yunnan, Xinjiang, Shanxi, Shanxi, and Sichuan have almost 0. Combined with the point outdegree, it can be found that the point outdegree of the main

livestock-producing provinces is generally greater than the point indegree, i. e., the"trickle-down effect"is greater than the "Siphon effect", so it is evident that the main livestock-producing provinces are exporting more resources than they are attracting. The main reason for this is that on the one hand, the major livestock-producing provinces need to constantly export animal products to other provinces, on the other hand, due to the relatively low degree of concentration in the animal husbandry industry and the lack of technological innovation and other reasons that lead to the low productivity and economic efficiency of the major livestock-producing provinces, and are unable to attract a large influx of human capital and other factors. Shanghai, Beijing, Jiangsu and Zhejiang are the most economically and socially developed provinces in China, thus attracting a large influx of talent, capital, technology and other resources into the region.

The Closeness centrality [74]can reflect the extent to which the high development of animal husbandry industry in a province is not controlled by other regions. On average, the closeness

**Table 7. Spatial correlation network centrality analysis of high quality development of animal husbandry in Chinese provinces.**

| Year | 2010 | | | | 2022 | | | |
|---|---|---|---|---|---|---|---|---|
| Index | OutDegree | InDegree | Closeness | Betweenness | OutDegree | InDegree | Closeness | Betweenness |
| Anhui | 4(6) | 1(12) | 29. 95(9) | 0. 53(16) | 4(8) | 4(10) | 14. 08(12) | 0. 2(18) |
| Beijing | 1(9) | 1(12) | 45. 93(1) | 1. 19(14) | 2(9) | 25(2) | 45. 73(1) | 2. 37(9) |
| Fujian | 8(2) | 24(2) | 22. 78(14) | 1. 32(12) | 8(4) | 15(4) | 14. 64(7) | 8. 43(1) |
| Gansu | 7(3) | 1(12) | 5. 72(22) | 0. 03(19) | 10(1) | 2(12) | 6. 21(21) | 0. 92(13) |
| Guangdong | 8(2) | 6(8) | 30. 10(7) | 15. 70(1) | 9(2) | 11(5) | 14. 26(10) | 3. 85(6) |
| Guangxi | 6(4) | 4(10) | 21. 60(15) | 0. 36(17) | 9(3) | 3(11) | 13. 12(16) | 2. 28(10) |
| Guizhou | 9(1) | 6(8) | 21. 44(16) | 7. 18(3) | 9(3) | 4(10) | 13. 18(15) | 3. 13(8) |
| Hainan | 6(4) | 0(13) | 5. 65(24) | 0(20) | 6(6) | 0(14) | 5. 65(26) | 0(22) |
| Hebei | 2(8) | 21(4) | 29. 30(10) | 2. 26(11) | 2(9) | 7(7) | 30. 22(3) | 0. 34(16) |
| Henan | 7(3) | 15(5) | 30. 07(8) | 6. 43(4) | 7(5) | 8(6) | 14. 54(8) | 5. 97(4) |
| Heilongjiang | 7(3) | 2(11) | 5. 88(20) | 0(20) | 8(4) | 0(14) | 6. 01(22) | 0(22) |
| Hubei | 5(5) | 26(1) | 27. 85(12) | 1. 20(13) | 7(5) | 7(7) | 13. 74(13) | 1. 69(12) |
| Hunan | 6(4) | 24(3) | 25. 34(13) | 1. 17(15) | 7(5) | 6(8) | 13. 26(14) | 0. 4(14) |
| Jilin | 7(3) | 2(11) | 5. 88(21) | 0(20) | 6(6) | 0(14) | 5. 52(28) | 0(22) |
| Jiangsu | 6(4) | 5(9) | 42. 85(3) | 6. 27(6) | 6(6) | 25(2) | 16. 05(5) | 6. 93(3) |
| Jiangxi | 6(4) | 2(11) | 28. 28(11) | 4. 42(8) | 6(6) | 5(9) | 14. 28(9) | 5. 08(5) |
| Liaoning | 4(6) | 14(6) | 5. 64(28) | 0(20) | 4(8) | 0(14) | 5. 50(29) | 0(22) |
| Inner Mongolia | 4(6) | 0(13) | 5. 52(30) | 0(20) | 4(8) | 1(13) | 5. 56(27) | 0. 02(21) |
| Ningxia | 7(3) | 0(13) | 6. 00(19) | 0(20) | 8(4) | 0(14) | 6. 45(19) | 0(22) |
| Qinghai | 7(3) | 4(10) | 5. 66(23) | 0(20) | 8(4) | 0(14) | 6. 44(20) | 0(22) |
| Shandong | 7(3) | 6(8) | 32. 66(5) | 6. 31(5) | 5(7) | 5(9) | 14. 16(11) | 0. 38(15) |
| Shanxi | 6(4) | 0(13) | 5. 55(29) | 0(20) | 4(8) | 0(14) | 5. 50(30) | 0(22) |
| Shanxi | 6(4) | 0(13) | 5. 65(25) | 0(20) | 5(7) | 2(12) | 5. 77(23) | 0. 09(20) |
| Shanghai | 7(3) | 0(13) | 45. 11(2) | 8. 08(2) | 7(5) | 26(1) | 16. 17(4) | 7. 83(2) |
| Sichuan | 6(4) | 0(13) | 14. 38(18) | 0. 14(18) | 8(4) | 1(13) | 11. 23(18) | 0. 1(19) |
| Tianjin | 3(7) | 0(13) | 31. 42(6) | 4. 38(9) | 2(9) | 11(5) | 31. 38(2) | 0. 22(17) |
| Xinjiang | 6(4) | 9(7) | 5. 65(26) | 0(20) | 7(5) | 0(14) | 5. 66(25) | 0(22) |
| Yunnan | 6(4) | 0(13) | 5. 65(27) | 0(20) | 8(4) | 0(14) | 5. 72(24) | 0(22) |
| Zhejiang | 6(4) | 0(13) | 37. 38(4) | 6. 04(7) | 5(7) | 16(3) | 15. 15(6) | 1. 89(11) |
| Chongqing | 7(3) | 4(10) | 16. 93(17) | 3. 46(10) | 8(4) | 5(9) | 12. 34(17) | 3. 54(7) |

Note:Rankings of provinces are in parentheses.

index of each province grows from 12. 92 in 2010 to 20. 06 in 2022, reflecting the increasing independence of the level of high-quality development of China's animal husbandry industry in each province and the increasing ability to control resources. The closeness indexes of Beijing, Shanghai, Tianjin, Jiangsu, Zhejiang and Guangdong are always in the top ten, reflecting the their low degree of external dependence and their ability to rely on the resources of their own provinces to achieve high-quality development of the livestock husbandry industry. The closeness index of Ningxia, Qinghai, Gansu, Heilongjiang, Shanxi, Yunnan, Xinjiang, Hainan, Inner Mongolia, Jilin, Liaoning and Shanxi is always below 10, reflecting that the main livestock- producing provinces in China have a weak independence of animal husbandry development, have little control over resources and information, and need the support of other provinces in terms of technological innovation, human capital, and management experience in the process of promoting high-quality development of the animal husbandry industry.

The degree of Betweenness centrality [74] can reflect the ability of each province to control the resource elements in the network of high-quality development of the animal husbandry industry. In 2010 and 2022, the betweenness centrality of Guangdong, Shanghai, Guizhou, Henan, Jiangsu and Jiangxi are consistently in the top ten, reflecting the fact that these provinces have a certain degree of control over the resource elements of the national animal husbandry development. The betweenness centrality of Shandong, Zhejiang, Tianjin and Hebei shows a clear downward trend, reflecting that the control of these five provinces over the resources of animal husbandry tends to decline. The betweenness centrality degree of Fujian shows a clear upward trend, reflecting the reinforcement of Fujian's control over resources and information in the high- quality network linkages of the animal husbandry industry. The betweenness centrality of Hunan, Hubei, Beijing, Sichuan, Gansu, Heilongjiang, Jilin, Liaoning, Ningxia, Qinghai, Hainan, Xinjiang, Inner Mongolia, Shanxi, Shanxi, and Yunnan are in a state of very low level, staying below 1, reflecting once again the weak control of resource elements in China's main farming producing regions. The main farming provinces have not yet developed animal husbandry into an absolutely dominant industry, market competitiveness is weak, and the phenomenon of resource outflow is obvious. The deep- seated reason lies in the fact that China's animal husbandry is still a decentralised operation, with a short industrial chain and low economic benefits for farmers, who are unable to give full play to the production benefits brought about by the agglomeration of resources [8,9,12].

## 3. 5. Analysis of factors affecting the high-quality development of China's animal husbandry industry

Table 8 reports the regression results of the factors influencing the level of high-quality development in China's animal husbandry industry. To ensure robust findings, this paper employs six regression methods. The initial column displays the results of the panel mixed least squares regression utilising clustered robust standard errors, while the second column presents the results of the fixed effects regression, also employing clustered robust standard errors. The third column details the results of the two-way fixed effects regression, controlling for both time and individual variables. The fourth column indicates the results of the random effects regression utilizing clustered robust standard errors. Specifically, both panel fixed-effects and random- effects models operate under the assumption of homoskedasticity and an absence of autocorrelation. Should heteroskedasticity and autocorrelation be present in the model, pseudo- regression may arise. This paper utilises both the White test and the BG test to appraise the presence of heteroskedasticity and autocorrelation in the model, respectively. With both tests rejecting the original hypothesis, the model constructed in this paper significantly exhibits both heteroskedasticity and autocorrelation. Drawing upon the research

**Table 8. Regression results of factors influencing the high-quality development of China's animal husbandry industry.**

| Variables | (1) | (2) | (3) | (4) | (5) | (6) |
|---|---|---|---|---|---|---|
| | OLS_robust | FE_robust | FE_TW | RE_robust | FGLS | PCSE |
| PGDP | 0.1110*** | 0.0130 | 0.0058 | 0.0096 | -0.0246*** | 0.0052 |
| | (-0.0312) | (-0.0200) | (0.0202) | (-0.0250) | (-0.0092) | (-0.0290) |
| URB | -0.0683 | 0.1520** | 0.3260*** | 0.1600*** | 0.3520*** | 0.3380*** |
| | (-0.0569) | (-0.0639) | (0.0856) | (-0.0543) | (-0.0285) | (-0.0628) |
| TEC | 0.0392 | 0.0393* | 0.0250*** | 0.0370* | 0.0189*** | 0.0284*** |
| | (-0.0253) | (-0.0197) | (0.0081) | (-0.0191) | (-0.0035) | (-0.0058) |
| TRA | 0.0189 | 0.0313** | 0.0263* | 0.0263* | 0.0182* | 0.0281*** |
| | (-0.0175) | (-0.0147) | (0.0146) | (-0.0157) | (-0.0097) | (-0.0085) |
| IND | 0.2170* | 0.2210 | 0.1910* | 0.2200*** | 0.0487 | 0.2020*** |
| | (-0.1140) | (-0.1340) | (0.1140) | (-0.0827) | (-0.0974) | (-0.0436) |
| FIN | -0.0065 | 0.0067 | 0.0066** | 0.0069* | 0.0019* | 0.0065*** |
| | (-0.0052) | (-0.0042) | (0.0026) | (-0.0037) | (-0.0010) | (-0.0025) |
| ARI | 0.0447 | 0.1330*** | 0.0878** | 0.0820** | 0.1050*** | 0.0946*** |
| | (-0.0609) | (-0.0382) | (0.0383) | (-0.0378) | (-0.0271) | (-0.0288) |
| POD | 0.0380*** | 0.0822 | 0.2170*** | 0.0300*** | 0.1680*** | 0.2360** |
| | (-0.0104) | (-0.1240) | (0.0662) | (-0.0111) | (-0.0377) | (-0.1090) |
| PFA | 0.5310*** | 0.0284 | 0.0073 | 0.1350 | 0.1020 | 0.0436 |
| | (-0.1870) | (-0.0920) | (0.1040) | (-0.0942) | (-0.0691) | (-0.0512) |
| Constant | 0.0044 | -0.0862 | -0.2160*** | -0.0596* | -0.1390*** | 4.1440 |
| | (-0.0429) | (-0.0630) | (0.0639) | (-0.0307) | (-0.0282) | (-3.3710) |
| Observations | 369 | 369 | 390 | 390 | 390 | 390 |
| R-squared | 0.572 | 0.567 | 0.515 | 0.479 | —— | 0.8734 |

Note:***represents p<0.01

**represents p<0.05

*represents p<0.1. Numbers in parentheses are standard deviations.

conducted by Xiaojun Yang and Li Cheng(2019) [75,76], two treatment methods are employed. Firstly, regression utilizing Feasible Generalised Least Squares (FGLS) is implemented, followed by the correction of the standard deviation of the Ordinary Least Squares (OLS) model regression utilizing the panel-corrected standard deviation (PCSE). The results of these regressions are presented in columns five and six of Table 6, respectively.

Regression analysis indicates that the level of high-quality development in China's animal husbandry industry is influenced by a multitude of factors. Most specifically, the rate of urbanisation demonstrates a significant positive correlation with this development, indicating that increased urbanisation actively cultivates improvement in the industry. This phenomenon can be attributed to several key factors. Firstly, rising urbanisation rates often lead to an increase in non-agricultural employment opportunities for farmers, contributing to a general upward trend in societal income levels [77]. Therefore, demand for livestock products experiences gradual growth, further stimulating the development of animal husbandry practices. Secondly, increased urbanisation facilitates improvements in human capital, attracting more skilled individuals to the field and thereby cultivating advancements in scientific and technological progress, as well as elevating management standards in the industry [78]. Thirdly, a greater level of urbanisation inevitably affects the animal husbandry industry's mode of production, leading to a decline in small-scale family farms and encouraging a shift towards larger-scale operations characterised by specialisation and modernisation. In conclusion, the increase of urbanisation

in China significantly generates far-reaching and positive consequences for the high-quality development of the nation's animal husbandry industry.

The scientific and technological advancement has a significantly positive effect on the progress of the animal husbandry industry. The Scientific and technological innovation should be regarded as the driving force behind such progress. Considering finite resources, it is a vital tool with which to optimise feed composition, maximise the use of waste products, optimize breeding programmes for superior livestock, and develop more sophisticated approaches to disease prevention and control. These factors, accordingly, lead to improved production efficiency, enhanced product quality and greater environmental friendliness, all of which contribute to a more sustainable future for the animal husbandry industry.

The quality of transport infrastructure significantly affects the progress of the animal husbandry industry towards high-quality development. Improved transport infrastructure is conducive to optimising the circulation and accessibility of livestock products in the market. This optimisation reduces transport costs while simultaneously preserving the freshness and quality of livestock products [79]. Moreover, it facilitates a more efficient and reliable system for the supply of essential production factors, such as feed and manufacturing equipment, required by livestock breeders. The enhancement of transport infrastructure also strengthens the animal husbandry industry chain by enhancing processing, storage, and transport capabilities for livestock products. Such improvements contribute to the increased value of livestock products and further extend the industry chain [80]. Simultaneously, improvements in transport infrastructure can propel the progress of cold chain logistics and e-commerce platforms for livestock products, thereby expanding both sales channels and market reach.

The share of the agricultural industry in the total industry can have a positive impact on the high quality of the animal husbandry sector. On the one hand, as the agricultural industry expands, the cultivation and supply of feed crop will also increase, which will help to increase the source of forage for the animal husbandry industry and reduce forage costs, thereby promoting the high-quality development of the animal husbandry industry. The development of the agricultural industry is also conducive to the combination of agriculture and livestock and recycling. The development of the agricultural industry can provide more agricultural and sideline products, which can be used as feed for the animal husbandry industry, thus promoting the combination of agriculture and livestock and the recycling of resources, thereby improving the efficiency and sustainability of the animal husbandry industry. On the other hand, the development of agricultural industry is usually accompanied by technical support and industrial coordination. Technological progress and industrial coordination can promote the synergistic development of agriculture and animal husbandry, forming a virtuous circle and promoting the development of animal husbandry to high quality.

The extent of agricultural financial development significantly contributes to the high-quality development of the animal husbandry industry. A sophisticated agricultural financial system, equipped to offer more financial support to the animal husbandry industry, is also better positioned to offer insurance products and risk management strategies, enhancing the industry's resilience against unforeseen events such as natural disasters and epidemics [81]. In addition, environmentally-focused financial instruments, including green credit and carbon finance, can steer animal husbandry production towards sustainable practices that conserve resources and protect the environment [82]. Finally, widespread access to financial services, particularly for smallholder farmers, can facilitate the modernisation of animal husbandry industry production methods, finally driving high-quality development throughout the industry.

The agricultural industry's structure significantly affects the high-quality development of the animal husbandry industry. Specifically, a greater proportion of animal husbandry industry output value in the overall agricultural output value positively correlates with the

optimisation of the animal husbandry industry. A higher proportional value signifies that animal husbandry production holds greater significance in the agricultural sector and experiences more robust consumer demand. This results in the allocation of increased resources (e. g., water resources, land resources, feedstuffs, etc.) towards the animal husbandry's development. Moreover, this significance attracts scientific research organisations, technological support, and innovative resources to be invested in animal husbandry. The governmental bodies are likely to introduce more supportive policies and funding to steer the animal husbandry industry towards a more efficient and sustainable model.

A certain correlation exists between population density and the level of high-quality development in the animal husbandry industry:a higher population density suggests conditions more conducive to the high-quality development of animal husbandry industry. Theoretically, a densely populated region implies a more extensive consumer market, yielding greater economic advantages for those engaged in animal husbandry industry. In addition, areas with high population density often exhibit a greater concentration of research facilities, technical expertise, and resources for innovation, cultivating a more fertile ground for technological advancement in animal husbandry industry [83]. The animal husbandry industry in such regions may also have a greater capacity to attract investment and technical assistance, thereby promoting more efficient and sustainable practices [84]. In conclusion, a higher population density generally signifies greater regional strength, which strengthens the capacity for high-quality development in the animal husbandry industry.

In order to further examine the regional heterogeneity of factors affecting the high-quality development of China's animal husbandry industry, regressions were conducted on the factors affecting the high-quality development of the animal husbandry industry in the eastern, central, and western regions, respectively. The results are shown in Table 9.

1. GDP per capita has had a positive impact on the high quality development of the animal husbandry in the eastern and western regions. However, for the central region, the rapid economic development has inhibited the high-quality development of the animal husbandry industry, possibly because with rapid economic development, the demand for water, land and other resources in the central region increased and a large amount of land was used for urban and industrialised construction, triggering a more serious competition for resources, which led to the animal husbandry industry facing a serious shortage of water and land resources and restrict the high-quality development of the animal husbandry industry. (2) The urbanisation rate had a significant positive impact on the high quality development of the animal husbandry in the eastern, western and the central regions and the coefficients are higher. It can be seen that with the increase in the level of urbanisation, the living standard of the residents has also been improved, and at the same time, the increase in urbanisation has had a positive impact on the large-scale operation of animal husbandry, which in turn promotes the high-quality development of the animal husbandry industry. (3) The level of scientific and technological development has a significant positive impact on the high-quality development of animal husbandry in the east, west and central regions of China. (4) The improved level of transport has had some positive impact on the eastern and central regions and no impact on the western region. The likely reason for this is that the good transport network in the eastern and central regions provides convenient marketing channels for animal products, promotes the circulation and allocation of factors of production, and improves transport efficiency and profitability, which in turn promotes high-quality development of the animal husbandry industry. The foundation for the development of the animal husbandry industry in the western region is relatively weak, and even if the level of transport is raised, it is difficult to achieve significant development due to

**Table 9. Heterogeneity regression results of factors influencing the high quality development of China's animal husbandry industry.**

| Variables | eastern | | western | | centre | |
|---|---|---|---|---|---|---|
| | FE_TW | PCSE | FE_TW | PCSE | FE_TW | PCSE |
| PGDP | 1. 2831*** | 0. 0324 | 0. 1560** | 0. 1450*** | -0. 7293 | -0. 2853** |
| | (0. 3454) | (0. 0225) | (0. 0697) | (0. 0494) | (0. 0700) | (0. 1427) |
| URB | 1. 3534* | 0. 2870*** | 0. 6890* | 0. 8300*** | 0. 5101* | 1. 7112*** |
| | (0. 6278) | (0. 0721) | (0. 3870) | (0. 3040) | (0. 2627) | (0. 4320) |
| TEC | -0. 1157 | 0. 0358*** | 0. 0380*** | 0. 0430*** | 0. 0427*** | 0. 1598*** |
| | (0. 1230) | (0. 0122) | (0. 0103) | (0. 0096) | (0. 0108) | (0. 0267) |
| TRA | -0. 1215 | 1. 0582*** | -0. 0141 | -0. 0077 | 0. 3491** | 0. 0526 |
| | (0. 3566) | (0. 0580) | (0. 0158) | (0. 0104) | (0. 1760) | (0. 1685) |
| IND | 0. 3797* | 0. 9014*** | 0. 5080 | 0. 6250** | 0. 1449 | 0. 4760*** |
| | (0. 1962) | (0. 2982) | (0. 3300) | (0. 2590) | (0. 0807) | (0. 0515) |
| FIN | 0. 1336*** | 0. 0133*** | 0. 0205*** | 0. 0223*** | 0. 0095 | -0. 0233 |
| | (0. 0334) | (0. 0049) | (0. 0049) | (0. 0051) | (0. 0102) | (0. 0265) |
| ARI | -0. 2272 | -0. 1089** | 0. 3810*** | 0. 3980*** | 0. 1849 | 0. 0217 |
| | (0. 1299) | (0. 0464) | (0. 0705) | (0. 0723) | (0. 1368) | (0. 0776) |
| POD | 0. 2177* | 0. 0542*** | 2. 4860*** | 2. 4220*** | 2. 4248*** | 0. 3968*** |
| | (0. 1063) | (0. 0097) | (0. 3910) | (0. 4060) | (0. 3185) | (0. 1456) |
| PFA | 0. 0549 | 0. 4975 | 0. 2210 | 0. 2440* | 0. 0819 | -0. 0054 |
| | (1. 0000) | (0. 4139) | (0. 1910) | (0. 1340) | (0. 0857) | (0. 1374) |

Note:***represents p<0. 01

**represents p<0. 05

*represents p<0. 1. Numbers in parentheses are standard deviations.

factors such as infrastructure and resource endowment. (5) The relationship between industrial structure and the high-quality development of the animal husbandry industry is significantly positive in the east, centre and west. This reflects the fact that the higher the proportion of agriculture in the three industries, the more favourable it is to the high-quality development of the animal husbandry industry, whether in the east, the centre or the west.. (6) There is a significant positive effect of the level of agricultural financial development on the high quality development of the animal husbandry sector in the eastern and western regions. The impact on the level of high-quality development of the animal husbandry industry in the central region is not significant, the possible reason is that agricultural finance in the central region is more oriented towards the plantation industry and has less coverage of the animal husbandry industry. The central region should promote more financial products and service support for the high-quality development of the animal husbandry industry, and formulate policies to attract more funds to invest in the high-quality construction of the animal husbandry industry. (7) The structure of the agricultural industry shows a positive correlation with the high-quality development of the animal husbandry industry in the central and western regions. It is negatively correlated with the high quality development of the animal husbandry industry in the eastern region, possibly because of the relative scarcity of water and land resources in the eastern region, and a high proportion of the value of animal husbandry output will exacerbate the resource constraints and limit the high quality of animal husbandry development. (8)Population density has a positive impact on the high-quality development of the livestock sector in the eastern, central and western regions. This reaffirms the fact that higher population densities are favourable to

stimulating the development of high quality in the animal husbandry sector. (9)Fiscal support for agriculture will have some impact on the western regions and no impact on the eastern and central regions. Shanxi and Henan are the main livestock breeding provinces in China, so the Government should provide more financial funds for the development of the animal husbandry industry in the central region and give full play to the role of financial funds in the high-quality development of the animal husbandry industry in the centre.

## 4. Conclusions and discussion

### 4. 1. Conclusions

Firstly, the level of high-quality development in China's animal husbandry industry exhibits a trend of slow growth. When categorised by dimension, this growth is reflected as follows:the level of scientific and technological &management surpasses that of environmental friendliness, which exceeds resource conservation. Output efficiency and, finally, product safety follow. Regionally, Shanghai and Jiangsu stand out as high-quality animal husbandry development areas. Five provinces–Anhui, Beijing, Fujian, Yunnan, and Chongqing–comprise the medium-quality tier. Eleven provinces, specifically Gansu, Guangdong, Guangxi, Guizhou, Hainan, Henan, Heilongjiang, Hubei, Inner Mongolia, Ningxia, and Qinghai, constitute the medium-low quality group. The remaining twelve provinces, Hebei, Hunan, Jilin, Jiangxi, Liaoning, Shandong, Shanxi, Shan xi, Sichuan, Tianjin, Xinjiang, and Zhejiang, represent the low-quality tier.

Secondly, regional differences in the level of high-quality development in China's animal husbandry industry are minimal and demonstrate a slight contraction. Intra- regional differences adhere to an East > Central > West pattern, while inter-regional differences follow an East and Central > East and West > Central and West. The contribution of these differences to the development has transitioned from an inter-region > intra-region > hypervariable density pattern in 2010 to a hypervariable density > intra-region > inter-region pattern in 2022. The effect of intra-regional differences on overall differences remains relatively stable. However, the effect of inter-regional differences on overall differences is decreasing, while the effect of hypervariable density is rising in significance.

Thirdly, from the temporal dynamic evolution trend, the level of high-quality development of the animal husbandry sector has increased in the eastern, central and western regions. The probability of each region maintaining its current level of high-quality animal husbandry development is high. Advancements in high-quality animal husbandry development primarily occur between adjacent levels, with cross-stage shifts being infrequent. The level of high-quality development in neighbouring areas significantly affects local development, reflecting the adage, 'One who is near vermilion is stained red, one who is near ink is stained black. 'The probability of upward transition is highest when a low-level area borders a medium-low level area, when a medium-low level area neighbours a medium-level area, and when a medium-level area is adjacent to a high-level area.

Fourth, the spatial network linkage level of China's high-quality development level of animal husbandry is low, and the hierarchical characteristics are obvious. The"trickle-down effect"is greater than the"Siphon effect"in major livestock-producing provinces such as Heilongjiang, Jilin, Liaoning, Inner Mongolia, Xinjiang, Shanxi, Shaanxi, Sichuan, Gansu and Yunnan, while the" Siphon effect"is greater than the"trickle- down effect"in economically and socially developing provinces such as Beijing, Shanghai, Jiangsu and Zhejiang.

Fifthly, the level of high-quality development in the animal husbandry industry is subject to the effect of several factors, including urbanisation rate, the level of scientific and technological

innovation, the level of transport infrastructure, industrial structure, the level of agricultural financial development, agricultural industry structure, and population density. Higher levels of urbanisation, scientific and technological innovation, transport infrastructure, agricultural financial development, and population density all correlate with higher levels of high-quality development in the animal husbandry industry. Moreover, a greater agricultural share in the industrial structure and a higher proportion of livestock in agriculture are also conducive to promoting high-quality development in the animal husbandry industry.

## 4. 2. Discussion

Based on the above research conclusions, this paper argues that:

1. Priority will be given to promoting the improvement of production efficiency and resource utilisation in the animal husbandry industry (This recommendation is based on the results of the five dimensions of the level of high-quality development of China's animal husbandry industry from 2010 to 2022). Specifically, there are the following aspects:On the one hand, concerning livestock and poultry germplasm resources, the resource consumption of Chinese livestock and poultry breeds is relatively higher than that of foreign livestock and poultry breeds. Therefore, the state should prioritise the enhancement of breeding capabilities and allocate increased resources towards the research of superior livestock and poultry breeds. Cultivating the growth of breeding enterprises, professional associations, and cooperatives, as well as other breeding-focused economic entities, is crucial. On the other hand, in terms of the breeding mode, the moderate scale operation of livestock and poultry breeding should be explored. On this basis, it will vigorously develop intelligent animal husbandry systems and promote intelligent animal husbandry production, integrated sales and business networks, and digital monitoring and management. It is necessary to increase infrastructure investment and promote the application of advanced equipment and facilities, such as breeding environment monitoring systems, physiological signs monitoring devices, automatic feeding systems and automatic waste treatment systems. Through these measures, the level of mechanisation and automation in the animal husbandry industry will be significantly improved.

2. Strengthening the regulation of livestock-related products (policy recommendations based on the low level of product safety in the five dimensions of the level of high-quality development of the animal husbandry sector). Establish mature quality and safety standards for animal products, testing norms and farm operation norms. Conduct regular inspections of farms to ensure that they meet safety and hygiene standards. Conduct regular vaccination and health checks on livestock and poultry to prevent the production and spread of pathogens. Conduct regular inspections of animal products to ensure that they meet safety standards. Establish a traceability system for animal products, which can be traced at every stage from breeding, processing to sales, to ensure the traceability of the products. Promote ecological farming models to reduce the use of antibiotics and chemicals, and reduce the impact on the environment and human health. Align with international standards and learn from advanced international experience in animal product safety management.

3. Strengthening the development of provincial linkages and the regional spatial spillover effect. There are fewer links between provinces for the high-quality development of China's livestock industry, so it is important to strengthen exchanges between provinces for the high-quality development of the livestock industry and encourage the sharing of resources. Jiangsu and Shanghai have a high level of high-quality livestock development; therefore, other provinces should learn from the experience of livestock management in both Jiangsu

and Shanghai. The Government needs to promote the flow of livestock-related resources to less efficient areas through policy guidance and market-based instruments, so as to improve the overall efficiency of resource utilisation. Establishing an information-sharing platform for the development of the animal husbandry industry and issuing timely policies, market information and technological advances in the development of the animal husbandry industry. It promotes the circulation and sharing of information between provinces and helps localities to scientifically plan the development of the livestock industry in accordance with market demand and resource conditions.

4. Focus on promoting the high-quality development of the animal husbandry industry in the main livestock-producing provinces (proposed in the light of the reality that the level of high-quality development of the animal husbandry industry in the main livestock-producing provinces is low). The main livestock-producing provinces should optimise policies for the introduction of talents and provide competitive salaries and benefits such as comprehensive social security, medical insurance, housing subsidies and other benefits. Establishment of standardised and large-scale breeding demonstration bases, animal husbandry science and technology demonstration bases, and animal husbandry industrial parks in the main livestock breeding provinces. The central Government should provide the main livestock- producing provinces with substantial financial allocations, set up special funds for the development of the animal husbandry industry, and provide more support for universities and research institutes in the main livestock-producing provinces. Simplify the market access and approval process related to animal husbandry, lower the operational threshold for enterprises, and encourage more enterprises to participate.

5. Coordinate the promotion of new urbanisation and the development of rural finance (policy recommendations based on factors affecting the level of high-quality development of the animal husbandry industry). Attracting the peasant population to work in the cities while controlling land occupation. Strengthening compulsory education in rural areas, in particular by vigorously developing vocational and technical education and training, so that the majority of the new generation of labourers in rural areas can acquire certain labour skills or expertise and be able to settle down in towns and cities. The government should formulate relevant policies and provide financial institutions with financial support, and encourage financial institutions to introduce financial products specifically designed for the high-quality development of the animal husbandry industry. Lower the financing threshold of the main body of farming and improve the accessibility of livestock business entities. Provide livestock insurance, futures and other financial tools to help herders avoid the risks associated with natural disasters and market fluctuations.

## Author Contributions

**Conceptualization:** Cuixia Li.

**Data curation:** Tiantian Su.

**Formal analysis:** Tiantian Su.

**Investigation:** Tiantian Su.

**Methodology:** Tiantian Su.

**Project administration:** Cuixia Li.

**Software:** Tiantian Su.

**Supervision:** Cuixia Li.

**Visualization:** Cuixia Li.

**Writing – original draft:** Tiantian Su.

**Writing – review & editing:** Cuixia Li.

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
