## [Decision Letter · Decision Letter 0]

17 Sep 2024

PONE-D-24-35886Spatial-Temporal Characteristics and Influence Factors of high-quality development of animal husbandry industry in ChinaPLOS ONE

Dear Dr. Su,

Thank you for submitting your manuscript to PLOS ONE. After careful consideration, we feel that it has merit but does not fully meet PLOS ONE’s publication criteria as it currently stands. Therefore, we invite you to submit a revised version of the manuscript that addresses the points raised during the review process.

We look forward to receiving your revised manuscript.

Kind regards,

Trung Quang Nguyen

Academic Editor

PLOS ONE

“This work was funded by the National Natural Science Foundation of China (The effect of infant milk powder safety trust index on product competitiveness—Index measurement, Correlation model construction and market simulation, Project number 71673042).”

3. In the online submission form you indicate that your data is not available for proprietary reasons and have provided a contact point for accessing this data. Please note that your current contact point is a co-author on this manuscript. According to our Data Policy, the contact point must not be an author on the manuscript and must be an institutional contact, ideally not an individual. Please revise your data statement to a non-author institutional point of contact, such as a data access or ethics committee, and send this to us via return email. Please also include contact information for the third party organization, and please include the full citation of where the data can be found.

Reviewers' comments:

Reviewer's Responses to Questions

**Comments to the Author**

1. Is the manuscript technically sound, and do the data support the conclusions?

Reviewer #1: Yes

Reviewer #2: Partly

Reviewer #3: Yes

Reviewer #4: Yes

Reviewer #5: No

2. Has the statistical analysis been performed appropriately and rigorously? 

Reviewer #1: Yes

Reviewer #2: Yes

Reviewer #3: Yes

Reviewer #4: Yes

Reviewer #5: No

3. Have the authors made all data underlying the findings in their manuscript fully available?

Reviewer #1: Yes

Reviewer #2: Yes

Reviewer #3: Yes

Reviewer #4: Yes

Reviewer #5: No

4. Is the manuscript presented in an intelligible fashion and written in standard English?

Reviewer #1: Yes

Reviewer #2: Yes

Reviewer #3: Yes

Reviewer #4: Yes

Reviewer #5: No

5. Review Comments to the Author

Reviewer #1: Areas for Improvement:

1. Introduction: The current introduction primarily focuses on the research background and existing literature, but it is somewhat weak in explaining the theoretical framework. It is recommended to add a section in the introduction specifically to elaborate on the theoretical framework of this study. This framework should address the following:

o Why were these five dimensions (output efficiency, product safety, resource conservation, environmental friendliness, and technological management level) chosen to evaluate the high-quality development of the livestock industry?

o How are these dimensions interrelated?

o How does this theoretical framework connect with existing theories of sustainable development or high-quality development? A clear theoretical framework will help readers better understand the logic and significance of the research.

2. Graph and Chart Optimization: Although the graphs and charts in the article provide a wealth of information, there is room for improvement:

o Figures 1 and 2: It is recommended to use different colors or patterns to distinguish the various indicators, enhancing readability. Additionally, consider adding trend lines to more clearly display the changes over time.

o Figures 4 and 5: These two figures contain a lot of information, which may confuse readers. It is suggested to split them into multiple sub-figures, with each sub-figure focusing on a specific region or comparison, to more clearly display the differences between regions.

o Tables 2 and 3: Consider using a heatmap to display the data, which would more intuitively show the differences and trends across different provinces.

3. Targeted Policy Recommendations: The current policy recommendations are somewhat broad and general; they could be made more targeted:

o For each major finding, provide corresponding specific policy recommendations. For example, based on the finding that the "technological management level" dimension scores the highest, suggest how to further enhance and promote advanced management practices.

o Propose differentiated development strategies based on the characteristics of different types of regions (high-quality, medium-quality, medium-low quality, and low-quality).

o Based on the results of the spatial correlation network analysis, suggest how to promote regional collaborative development.

o Discuss in detail how to promote the high-quality development of the livestock industry by adjusting these factors (such as urbanization rate, technological innovation level, transportation infrastructure) based on the results of the influencing factor analysis.

o Consider adding a timeline or roadmap that outlines how these policy recommendations should be implemented in phases. By making these improvements, the article's theoretical foundation will be more robust, the presentation of research results will be clearer and more intuitive, and the policy recommendations will be more actionable and targeted, thereby further enhancing the academic value and practical significance of the paper.

Grammar:

1. There are a few errors in the use of articles, for example:

o Original: "The livestock industry stands as a pillar of the agricultural sector."

o Suggested: "The livestock industry stands as a pillar of the agricultural sector."

2. Some sentences are overly complex and could be split:

o Original: "Driven by increasing demand for meat, eggs, milk, and other livestock products, China's livestock industry has become a cornerstone of its agricultural sector, representing 34% of agricultural output[1]."

o Suggested: "China's livestock industry has become a cornerstone of its agricultural sector, representing 34% of agricultural output[1]. This development has been driven by increasing demand for meat, eggs, milk, and other livestock products."

3. There are some inconsistencies in verb tense, for example:

o Original: "The entropy weight method hypothesizes that indicators with higher entropy values should be assigned greater weights, and vice versa."

o Suggested: "The entropy weight method hypothesizes that indicators with higher entropy values should be assigned greater weights, and vice versa."

Citation:

1. The citation format is generally correct, but some punctuation needs adjustment, for example:

o Original: "...[23-24]"

o Suggested: "...[23, 24]"

2. Some citations may need to be updated or supplemented:

o When discussing the latest policies (such as the 2023 revision of the "Animal Husbandry Law of the People's Republic of China"), it is recommended to directly cite official documents.

3. Some important points lack citation support, for example:

o "China's livestock farming is still a decentralized operation, with a short industrial chain and low economic benefits for farmers, who are unable to fully realize the production benefits brought by the agglomeration of resources." This point needs to be supported by relevant research or reports.

4. When discussing the development of the livestock industry in specific provinces, consider adding some local studies to strengthen the argument.

5. When introducing research methods (such as the entropy weight method, Dagum Gini coefficient, etc.), consider citing the original or classic application literature on these methods.

Overall, this is a high-quality research paper with significant theoretical and practical implications for understanding and promoting the high-quality development of China's livestock industry. It is recommended for major revision and subsequent acceptance for publication.

Reviewer #2: This paper evaluated the spatial and temporal characteristics of the quality of China's livestock development by constructing an evaluation system for the high-quality development of China’s livestock industry. The system was constructed with five principal dimensions: output efficiency, product safety, resource conservation, environmental friendliness, and the level of scientific and technological management. Moreover, the impact of the following factors on the high-quality development of China’s animal husbandry industry was investigated: GDP per capita, urbanization rate, level of scientific and technological development, level of transportation, industrial structure, level of agricultural financial development, structure of agricultural industry, population density, and financial support for agriculture. The findings indicated that the high-quality development of China’s animal husbandry industry demonstrates a notable degree of path dependence, with regional disparities exhibiting a tendency towards convergence and development in neighboring regions exerting a mutual influence. Moreover, the urbanization rate, the level of science and technology innovation, transportation infrastructure, the level of agricultural financial development, and population density were identified as exerting a positive influence on the high-quality development of China’s animal husbandry industry.

Overall, the structure of the paper was logical and the discussion and analysis were provided at a more comprehensive level. However, a few remaining issues would benefit from further revision and improvement.

1. Lines 743-771 presented a comprehensive description of the impact factor analysis model. It would appear more reasonable to reclassify them as “2.1 Research Methods.”

2. The description of the regions was inconsistent. For example, the country’s 30 provinces were divided into eastern, central, and western regions in lines 301-305, whereas them were described as eastern, central, western, and north-eastern regions in lines 886-887.

3. It seems more appropriate to rename the section “3. Results” as “3. Results and Discussion.”

4. A number of the statements lacked a clear and direct connection to the serial number of the chart, or were not adequately supported by data or literature.

(1) The text in lines 313-315, "The overall standard ... the period from 2010 to 2022," seemed to be related to Table 2. Similarly, the phrase "rising from 0.13 in 2010 to 0.17 in 2022" in line 316 appeared to correspond to Figure 2. The terms "level of scientific and technological management" in line 317 and "level of resource conservation" in line 319 lacked the serial numbers of the charts to which they referred. There were many similar issues that required a comprehensive review of the text and the addition of graphical serial numbers that corresponded to the textual descriptions.

(2) Some arguments need more supporting data or literature, such as “has remained stable but at a lower level” (line 324), “Shanghai, an international metropolis...enacted various environmental management measures” (lines 345-347), “Fujian exhibits...breeding livestock and poultry stock nationally” (lines 363-364), “Yunnan Province has...animal husbandry” (lines 373-375).

Similarly, these descriptions, “The previous study measured... its spatial and temporal evolution trends” (lines 743-744) and “Quality of transportation infrastructure...towards high-quality development” (lines 819-820), required substantiation with evidence from the relevant literature.

In conclusion, there were additional issues similar to those mentioned above that required further supplementation and improvement.

5. Some of the formatting of the text requires further standardization.

(1) There was a discrepancy in the formatting of the literature cited on lines 114 and 786 that did not align with the formatting of all other citations in the full text. The format on lines 114 and 786 was “Xiong Xuezhen (2022) [17]” and “Zhao Lei and Cheng Fang (2019)”, respectively.

(2) The format “during the observation period. utilizing 2010 as a baseline” (lines 445-446) would appear to be “during the observation period, utilizing 2010 as a baseline”.

(3) The wording of “firstly” in line 796, “moreover” in line 800, and “moreover” in line 803 appeared to be more appropriate when modified to “firstly, secondly, and thirdly.”

Reviewer #3: The paper constructs evaluation indicators for high quality animal husbandry development in the Chinese scenario and evaluated developments in Animal Husbandry industry in China. Adequate quantum of data, 12- year period, was evaluated. Based on the findings, the authors put forward many suggestions for animal husbandry development stressing on technological innovation, prioritizing enhancement of breeding capabilities & development of superior breeds, improving allocation for R&D, and encouraging collaborations, mechanization & automation and ecologically sound livestock operations. Agricultural reformation, changing policies and liberal financing for AH sector are also suggested. Recommendations put forward are general in nature though lack novelty. However, this scientific analysis further validated the general concepts which is the relevance of this paper. Additionally, the study put forward influencing factors on animal husbandry developments in China and findings can address issues in the current system making way for improvement in Animal Husbandry activities of China.

Paper is well written but needs improvement on readability. All sections including results and conclusions could be more concise limiting generalized statements and repetitions.

Reviewer #4: This is a well written manuscript the provides a deatiled insight in China's livestock industry. I enjoyed reading it and got many useful information from it. Scientifically, it is well structured and the methods that are used are sound. With minor revisions, I believe it will be appropriate for publication in Plos one journal. Please see the attached file for my suggestions.

Reviewer #5: The manuscript „Spatial-Temporal Characteristics and Influence Factors of high-quality development of animal husbandry industry in China“ is a complex modelling study that intends to study somehow the development of the animal husbandry industry in China. The manuscript is somehow hard to read and must be improved in language and structure. The methodology is to a large extend unclear and how it led to the results. Furthermore, are the results and the discussion mixed, which should be for readability be separated. The displays also require a major revision. For many parameters it remains open how they were obtained or calculated. In general, presentation must be improved and the aims be clarified. Language should be improved with a professional language service. Yet, presentation of aims, methods and results does not allow a full understanding of what have been conducted by the authors.

- Lines 8-30: A lot a free spaces are missing after comma and full stop

- Line 24: remove “,´” after adage

- Lines 24-25: I do not get that what you intend to say here?

- Line 29: What is GDP? Please introduce abbreviations upon first usage.

- Lines 78-79: Please provide a reference for this.

- Lines 79-80: Please provide a reference for this.

- Lines 80-82: Please provide a reference for this.

- Lines 82-83: Please provide a reference for this.

- Lines 91-97: I do not get the sentence completely, it is too long and to detailed. Please clarify. Further, what is intended to mean “background colour” in line 92.

- Lines 103-116: What do you intent to communicate here to the reader?

- Lines 123-124: Please cite the mentioned paper.

- Lines 123-130: The entire introduction is hard to follow, and here it is hard to get even an idea what the authors aim with this paper. I recommend that the authors should rewrite the entire introduction and the aims and make much clearer what their aims are.

- Lines 133-206: For what are these methods intended to use? I remains very unclear, also what data input is used etc.

- Lines 208-209: I suggest to remove political statements from the paper.

- Lines 212-213: Why these?

- Lines 207-297: This section provides many data, but needs much more references.

- Lines 298-306: Please provide links/reference to this data. It this data public available?

- Lines 312-422; 433-511; 526-562; 567-630; 631-735;742-959: The results are extremely hard to read and to understand and also which results are obtained by which method.

- Figure 1: y- and x-axis labels are missing. Furthermore, should the colour lines explained in the legend text.

- Figure 2: Axis labels are missing.

- Table 2: What are the numbers express in the table per year? How are the ranks estimated? Is the growth rate per 2year cycle calculated and averaged over the entire number of cycles or is it the annual mean when only considering 2010 and 2022?

- Table 3: Based on which criteria were the categories set, i.e. why is 0.25-0.31 high quality and why not another higher number? Why are the categories not same sizes, i.e. ranging from 0.04 to 0.06. Why are the highest two categories referred to livestock industry and the lower two to animal husbandry? What is “medium-low” ?

- -Table 4: What exactly do the numbers in the table say, which dimension/scale etc and how where they obtained?

- Figure 3: Remove grid please. And y-axis label is missing.

- Figure 4: y-axis label overlaps with scale. Please remove grid.

- Figure 5: remove grid please and add y-axis label.

- Figure 6: Please add axis labels.

- Line 528: The usage of this software should have been mentioned already in the methods.

- Table 5: Please explain table better, why are categories in both, i.e. rows and columns? And with respect to the categories, see my comment to Tabl. 3

- Table 7: What does numbers mean and what do numbers in brackets mean?

- Table 8+9: Data and abbreviations in the table are not properly explained, furthermore also indicate what stars and brackets mean, levels of significances?

- Line 959: No discussion is provided. However, the results section is mixed up with discussion parts. Please separate both sections clearly, that may help to make reading of the result section better readable.

6. PLOS authors have the option to publish the peer review history of their article (what does this mean?). If published, this will include your full peer review and any attached files.

Reviewer #1: **Yes: **FANGFANG GUO

Reviewer #2: No

Reviewer #3: No

Reviewer #4: **Yes: **Dr. George Symeon

Reviewer #5: No

---

## [Author Response · Author response to Decision Letter 0]

10 Oct 2024

We thank the reviewers for their comments. I feel that the quality of the article has been greatly improved after I revised it according to the reviewers' comments.

Point1:Introduction: The current introduction primarily focuses on the research background and existing literature, but it is somewhat weak in explaining the theoretical framework. It is recommended to add a section in the introduction specifically to elaborate on the theoretical framework of this study. This framework should address the following:

o Why were these five dimensions (output efficiency, product safety, resource conservation, environmental friendliness, and technological management level) chosen to evaluate the high-quality development of the livestock industry?

o How are these dimensions interrelated?

o How does this theoretical framework connect with existing theories of sustainable development or high-quality development? A clear theoretical framework will help readers better understand the logic and significance of the research.

Author response1:

The theoretical part of this paper mainly focuses on the construction of indicators in 2.2.

The reason why these five dimensions are used to construct the evaluation indicators for the level of high-quality development of the livestock industry is mainly due to the reference to the document issued by the General Office of the State Council of China in 2020 - Opinions of the General Office of the State Council on Promoting the High-Quality Development of the Animal Husbandry Industry, an authoritative document which explicitly puts forward the need to form a ‘new pattern of high-quality development that is highly efficient in terms of output, safe in terms of products, resource-saving, environmentally friendly, and effective in terms of regulation and control’. As the indicators related to ‘effective regulation’ are difficult to obtain, coupled with the fact that the level of science and technology ＆management are crucial to the high-quality development of the livestock industry, this paper selects indicators from both the level of science and technology and the level of management of the livestock industry. In turn, it constructs the evaluation index of the level of high-quality development of animal husbandry. Undeniably, due to the availability of data, the level of high-quality development of animal husbandry constructed in this paper is bound to have limitations, but in the construction of the indicators always follow the principles of scientific, comprehensive, systematic and complete construction of indicators, and try to be as close as possible to the reality of China's animal husbandry development, and therefore can also be a certain degree of response to China's high-quality level of development of the animal husbandry industry.

In the selection of specific indicators, the main reference was to existing research on the modernisation and green development of the livestock industry.

The four indicators of high output efficiency,product safety,resource conservation and environmental friendliness are parallel to each other.The science and technology ＆management levels are the internal driving forces that can promote the synergistic development of output efficiency, product safety, resource conservation and environmental friendliness. However, in this paper, this indicator of science and technology ＆management level is used as a supplementary indicator to evaluate the high-quality development level of animal husbandry together with output efficiency, product safety, resource conservation and environmental friendliness.

Point2:Graph and Chart Optimization: Although the graphs and charts in the article provide a wealth of information, there is room for improvement:

o Figures 1 and 2: It is recommended to use different colors or patterns to distinguish the various indicators, enhancing readability. Additionally, consider adding trend lines to more clearly display the changes over time.

Author response:Thanks to the reviewers for their comments. Figures 1 and 2 do have some problems and I have improved them. Figures 1 and 2 are line graphs with different colours used for each fold line. On top of that, trend lines were added and it is easy to see the trend of the different fold lines.

o Figures 4 and 5: These two figures contain a lot of information, which may confuse readers. It is suggested to split them into multiple sub-figures, with each sub-figure focusing on a specific region or comparison, to more clearly display the differences between regions.

Author response:Thanks to the reviewers for their comments. However, I think splitting it up would be a bit of a waste of space. In addition, putting the three lines of intra-regional differences and the three lines of inter-regional differences together would be more able to compare the intra-regional differences of East, Central and West and the inter-regional differences of East-Central, Central-West and East-West.

o Tables 2 and 3: Consider using a heatmap to display the data, which would more intuitively show the differences and trends across different provinces.

Author response:We thank the reviewers for their comments. I think these two tables are a clear presentation of what I want to say. For table 2, the heat map is not able to show the average annual growth rate. So here in this paper I still want to present it in the original table.

Point3:Targeted Policy Recommendations: The current policy recommendations are somewhat broad and general; they could be made more targeted:

o For each major finding, provide corresponding specific policy recommendations. For example, based on the finding that the "technological management level" dimension scores the highest, suggest how to further enhance and promote advanced management practices.(From this section, it can be seen that the production efficiency and resource utilisation efficiency of China's high-quality livestock development is still very low, therefore, the policy recommendation of this paper is to prioritise the promotion of the production efficiency of the livestock industry.The level of product safety is also low, hence the policy recommendations to ensure product safety.)

o Propose differentiated development strategies based on the characteristics of different types of regions (high-quality, medium-quality, medium-low quality, and low-quality).(From this part, it can be seen that the level of high-quality development of the livestock industry in China's main livestock- producing provinces is still very low; therefore, suggestions such as focusing on the main livestock-producing provinces and giving policy preferences to the main livestock-producing provinces are put forward.）

o Based on the results of the spatial correlation network analysis, suggest how to promote regional collaborative development(The policy recommendations for this section are to strengthen regional linkage development and to take advantage of the spatial spillover effects of developed provinces.).

o Discuss in detail how to promote the high-quality development of the livestock industry by adjusting these factors (such as urbanization rate, technological innovation level, transportation infrastructure) based on the results of the influencing factor analysis.(Promoting the development and application of science and technology has been highlighted in the one-location proposal. This focuses on the need to sustain urbanisation and rural financial development. Urbanisation development is conducive to the large-scale development of livestock farming, and rural financial development is conducive to providing financial support for high-quality development of the livestock sector.)

o Consider adding a timeline or roadmap that outlines how these policy recommendations should be implemented in phases. By making these improvements, the article's theoretical foundation will be more robust, the presentation of research results will be clearer and more intuitive, and the policy recommendations will be more actionable and targeted, thereby further enhancing the academic value and practical significance of the paper.

Author response: The policy recommendations section of the article is indeed shallow and lacks substance. Therefore, I have revised the policy recommendations of the article. The policy recommendations are mainly in the four areas of prioritising the promotion of livestock production efficiency, guaranteeing product safety, mainly promoting the level of high-quality development of livestock in the main livestock-producing provinces, promoting the increase in the urbanisation rate and the development of rural finance.

In terms of prioritising the promotion of livestock production efficiency, the main focus has been on accelerating the layout of good breed cultivation and promoting the development of production modes in the direction of scaling up and intelligentisation. In the protection of product safety, mainly from the regular vaccination of livestock and poultry and health checks; the establishment of livestock products traceability system and other aspects of the overflow of recommendations. In terms of focusing on the development of major livestock production areas, policy suggestions were made mainly in terms of the need for major livestock-producing provinces to attract human resources, and for the central government to provide more financial pens for major livestock-producing provinces. With regard to promoting the level of urbanisation and rural financial development, recommendations were made mainly in terms of strengthening compulsory education in rural areas and formulating financial products specifically designed for the high-quality development of the livestock industry.

Point4:Grammar:

1. There are a few errors in the use of articles, for example:

o Original: "The livestock industry stands as a pillar of the agricultural sector."

o Suggested: "The livestock industry stands as a pillar of the agricultural sector."

Author response:Thanks to the reviewers for their comments. It has been revised and the full text has been read through, with a focus on the correct use of articles' articles, and corrections have been made to incorrect use

2. Some sentences are overly complex and could be split:

o Original: "Driven by increasing demand for meat, eggs, milk, and other livestock products, China's livestock industry has become a cornerstone of its agricultural sector, representing 34% of agricultural output[1]."

o Suggested: "China's livestock industry has become a cornerstone of its agricultural sector, representing 34% of agricultural output[1]. This development has been driven by increasing demand for meat, eggs, milk, and other livestock products."

Author response:Thanks to the reviewers for their comments. The text does contain a large number of complex sentences. The text has been read through in detail and the complex sentences have been split and modified to make them easier to understand.

3. There are some inconsistencies in verb tense, for example:

o Original: "The entropy weight method hypothesizes that indicators with higher entropy values should be assigned greater weights, and vice versa."

o Suggested: "The entropy weight method hypothesizes that indicators with higher entropy values should be assigned greater weights, and vice versa."

Author response:Thanks to the reviewers for their comments. Revisions have been made.

Point5:Citation:

1. The citation format is generally correct, but some punctuation needs adjustment, for example:

o Original: "...[23-24]"

o Suggested: "...[23, 24]"

Author response5:Thanks to the reviewers for the heads up. The paper has been revised to cite two literature references in the text.

2. Some citations may need to be updated or supplemented:

o When discussing the latest policies (such as the 2023 revision of the "Animal Husbandry Law of the People's Republic of China"), it is recommended to directly cite official documents.

Author response:The citation in the text is the official document.

3. Some important points lack citation support, for example:

o "China's livestock farming is still a decentralized operation, with a short industrial chain and low economic benefits for farmers, who are unable to fully realize the production benefits brought by the agglomeration of resources." This point needs to be supported by relevant research or reports.

Author response:Already modified and cited

4. When discussing the development of the livestock industry in specific provinces, consider adding some local studies to strengthen the argument.

Author response:Thanks to the reviewer for the reminder. This paper does cite some literature in the section on province discussion in chapter 3.1, but due to my own negligence I forgot to add the references for this section. I have added the references.For example, the phrase ‘Shanghai,an international metropolis with leading scientific research institutions and a skilled workforce,prioritises environmental protection and has enacted various environmental management measures’ is largely a reference to the literature on ‘Virginie Arantes;Can Zou;Yue CheCope with waste: A government-NGO collaborative governance approach in Shanghai[J]. Journal of environmental management, 2020, 259: 109653’. The phrase”Jiangsu Province,characterised by its diverse animal husbandry models,centres its approach on scientific and technological advancements,industry upgrades,ecological farming practices,standardised management, and brand development.In addition,the province has implemented a suite of policies promoting livestock industry expansion,including financial subsidies,tax incentives,and scientific and technological support”is largely a reference to the literature on “Qi D. Transitions to Ecological Agriculture in Nan**g, China: Farm Types, Social-Political Networks, and Rural Communities[J].”Similarly,the phrase”Ningxia and Gansu face limitations in achieving a high level of high-quality development in animal husbandry due to factors such as the relative scarcity of water resources” is largely a reference to the literature on “Bin Liu;Feilian Zhang;Xiaosheng Qin;Zhe Wu;Xiaolan Wang;Yuanyuan He.Spatiotemporal assessment of water security in China: An integrated supply-demand coupling model[J].Journal of Cleaner Production, 2021, 321: 128955”.

Point6:When introducing research methods (such as the entropy weight method, Dagum Gini coefficient, etc.), consider citing the original or classic application literature on these methods.

Author response6:Thanks to the reviewers for the heads up. I have carefully studied the classical literature related to this paper with great benefit and have cited the classical literature at appropriate places. The specific citations can be seen in the research methods section of the text.

Overall, this is a high-quality research paper with significant theoretical and practical implications for understanding and promoting the high-quality development of China's livestock industry. It is recommended for major revision and subsequent acceptance for publication.

Reviewer #2: This paper evaluated the spatial and temporal characteristics of the quality of China's livestock development by constructing an evaluation system for the high-quality development of China’s livestock industry. The system was constructed with five principal dimensions: output efficiency, product safety, resource conservation, environmental friendliness, and the level of scientific and technological management. Moreover, the impact of the following factors on the high-quality development of China’s animal husbandry industry was investigated: GDP per capita, urbanization rate, level of scientific and technological development, level of transportation, industrial structure, level of agricultural financial development, structure of agricultural industry, population density, and financial support for agriculture. The findings indicated that the high-quality development of China’s animal husbandry industry demonstrates a notable degree of path dependence, with regional disparities exhibiting a tendency towards convergence and development in neighboring regions exerting a mutual influence. Moreover, the urbanization rate, the level of science a

---

## [Decision Letter · Decision Letter 1]

4 Nov 2024

Spatial-Temporal Characteristics and Influence Factors of

high-quality development of animal husbandry industry in China

PONE-D-24-35886R1

Dear Dr. Tiantian Su,

We’re pleased to inform you that your manuscript has been judged scientifically suitable for publication and will be formally accepted for publication once it meets all outstanding technical requirements.

Kind regards,

Trung Quang Nguyen

Academic Editor

PLOS ONE

Additional Editor Comments (optional):

Reviewers' comments:

Reviewer's Responses to Questions

**Comments to the Author**

1. If the authors have adequately addressed your comments raised in a previous round of review and you feel that this manuscript is now acceptable for publication, you may indicate that here to bypass the “Comments to the Author” section, enter your conflict of interest statement in the “Confidential to Editor” section, and submit your "Accept" recommendation.

Reviewer #1: All comments have been addressed

2. Is the manuscript technically sound, and do the data support the conclusions?

Reviewer #1: Yes

3. Has the statistical analysis been performed appropriately and rigorously? 

Reviewer #1: Yes

4. Have the authors made all data underlying the findings in their manuscript fully available?

Reviewer #1: Yes

5. Is the manuscript presented in an intelligible fashion and written in standard English?

Reviewer #1: Yes

6. Review Comments to the Author

Reviewer #1: I have carefully reviewed the revised manuscript titled "Spatial-Temporal Characteristics and Influence Factors of high-quality development of animal husbandry industry in China" for the second time. I am pleased to report that the authors have adequately addressed the concerns raised in the initial review and have made appropriate improvements to the paper.

After thorough examination, I find no significant grammatical errors or obvious mistakes in the revised version. The authors have successfully refined their work, enhancing its overall quality and clarity. The paper now presents a more cohesive and well-structured analysis of the high-quality development in China's animal husbandry industry.

The strengths of the manuscript, as noted in my previous review, remain intact:

A comprehensive evaluation framework for high-quality development in the animal husbandry sector.

Robust quantitative analysis using multiple methodologies.

Valuable insights into regional disparities, temporal trends, and spatial correlations.

Identification of key influencing factors with important policy implications.

Clear presentation of results with appropriate supporting visuals.

Given the thorough revisions and the absence of any significant issues, I recommend that this manuscript be accepted for publication in its current form. The paper makes a substantial contribution to the field and will be of great interest to researchers and policymakers working on sustainable agricultural development in China and beyond.

Should you require any further clarification, please do not hesitate to contact me.

7. PLOS authors have the option to publish the peer review history of their article (what does this mean?). If published, this will include your full peer review and any attached files.

Reviewer #1: No

---

## [Editor Report · Acceptance letter]

16 Dec 2024

PONE-D-24-35886R1 

PLOS ONE

Dear Dr. Su, 

I'm pleased to inform you that your manuscript has been deemed suitable for publication in PLOS ONE. Congratulations! Your manuscript is now being handed over to our production team.

Kind regards, 

on behalf of

Dr. Trung Quang Nguyen 

Academic Editor

PLOS ONE